# Functional correlates of clinical phenotype and severity in recurrent *SCN2A* variants

Géza Berecki [1,15✉], Katherine B. Howell[1,2,3,4,15], Jacqueline Heighway[1], Nelson Olivier[5], Jill Rodda[4], Isabella Overmars [4], Danique R. M. Vlaskamp[6], Tyson L. Ware[7], Simone Ardern-Holmes[8], Gaetan Lesca[9], Michael Alber[10], Pierangelo Veggiotti[11], Ingrid E. Scheffer[2,3,4,12], Samuel F. Berkovic [12], Markus Wolff[13] & Steven Petrou[1,5,14✉]

In *SCN2A*-related disorders, there is an urgent demand to establish efficient methods for determining the gain- (GoF) or loss-of-function (LoF) character of variants, to identify suitable candidates for precision therapies. Here we classify clinical phenotypes of 179 individuals with 38 recurrent *SCN2A* variants as early-infantile or later-onset epilepsy, or intellectual disability/autism spectrum disorder (ID/ASD) and assess the functional impact of 13 variants using dynamic action potential clamp (DAPC) and voltage clamp. Results show that 36/38 variants are associated with only one phenotypic group (30 early-infantile, 5 later-onset, 1 ID/ASD). Unexpectedly, we revealed major differences in outcome severity between individuals with the same variant for 40% of early-infantile variants studied. DAPC was superior to voltage clamp in predicting the impact of mutations on neuronal excitability and confirmed GoF produces early-infantile phenotypes and LoF later-onset phenotypes. For one early-infantile variant, the co-expression of the $\alpha_1$ and $\beta_2$ subunits of the $Na_v1.2$ channel was needed to unveil functional impact, confirming the prediction of 3D molecular modeling. Neither DAPC nor voltage clamp reliably predicted phenotypic severity of early-infantile variants. Genotype, phenotypic group and DAPC are accurate predictors of the biophysical impact of *SCN2A* variants, but other approaches are needed to predict severity.

[1] Ion Channels and Human Diseases Group, The Florey Institute of Neuroscience and Mental Health, University of Melbourne, Parkville, VIC 3052, Australia. [2] Department of Neurology, Royal Children's Hospital, Parkville, VIC 3052, Australia. [3] Department of Paediatrics, University of Melbourne, Parkville, VIC 3052, Australia. [4] Murdoch Children's Research Institute, Parkville, VIC 3052, Australia. [5] Praxis Precision Medicines, Inc, Cambridge, MA 02142, USA. [6] Departments of Neurology and Genetics, University Medical Center Groningen, University of Groningen, Groningen, The Netherlands. [7] Department of Paediatrics, Royal Hobart Hospital, Hobart, TAS 7000, Australia. [8] Department of Neurology, Children's Hospital Westmead, Sydney, NSW, Australia. [9] Service de Génétique, Hospices Civils de Lyon, 69002 Lyon, France. [10] Department of Pediatric Neurology and Developmental Medicine, University Children's Hospital, Tübingen, Germany. [11] Pediatric Neurology Unit, V. Buzzi Children's Hospital, Milan, Italy. [12] Epilepsy Research Centre, Department of Medicine, University of Melbourne, Austin Health, Heidelberg, VIC 3084, Australia. [13] Pediatric Neurology, Vivantes Hospital Neukölln, Berlin, Germany. [14] Department of the Florey Institute, University of Melbourne, Parkville, VIC 3050, Australia. [15] These authors contributed equally: Géza Berecki, Katherine B. Howell. ✉email: geza.berecki@florey.edu.au; steven.petrou@florey.edu.au

Mutations in the *SCN2A* gene encoding the voltage-gated sodium channel Na$_v$1.2 are one of the most frequent monogenic causes of neurodevelopmental disorders, with a number of distinct phenotypes reported[1–6]. Biophysical studies have suggested that gain-of-function (GoF) variants cause early-infantile epilepsies of variable severity, with seizure onset typically (but not exclusively) occurring before age 3 months and sodium channel blocking (SCB) anti-seizure medications (ASMs) reducing seizures[1,3]. In contrast, loss-of-function (LoF) variants result in either later-onset epilepsies (with seizures that may worsen with SCB ASMs), or ASD/ID without epilepsy (ASD/ID)[3,7,8].

While this correlation between genotype, biophysical impact and phenotypic group appears robust, for many of the >150 variants reported, clinical information is available for only a small number of individuals. In addition, functional testing failed to resolve the biophysical impact of some mutations[7,9]. The most useful platform for assessing the biophysical impact of *SCN2A* variants remain to be determined. Voltage clamp is a commonly used technique for assessing the biophysical properties of Na$_v$1.2 channels. However, interpretation of voltage clamp data can be difficult for variants exhibiting complex or opposing changes in their biophysical characteristics relative to the wild-type variant, as observed with a number of pathogenic *SCN2A* variants[7,9–12]. We have recently validated the dynamic action potential clamp (DAPC) technique for efficiently predicting the functional impact of missense *SCN2A* variants. In DAPC mode, a real-time coupling is established between a mammalian cell heterologously expressing a mutant Na$_v$1.2 channel and an excitatory neuron model incorporating virtual conductances. Such a hybrid neuron results in a unique action potential firing profile, depending on the biophysical characteristics of the implemented Na$_v$1.2 variant[7]. As we studied only four Na$_v$1.2 variants previously (wild-type, R1882Q, L1563V, R853Q), investigation of additional variants is needed to provide reassuring evidence that DAPC can consistently overcome the limitations of the voltage clamp approach.

Variation in severity is striking in early-infantile/GoF *SCN2A* disorders. At the milder end is S(F)NIS (self-limited (familial) neonatal-infantile seizures), characterized by treatment-responsive seizures that remit by the end of infancy with normal development[2,13]. More severe epilepsy phenotypes, also known as early-infantile developmental and epileptic encephalopathy (DEE), are associated with poor developmental outcomes and seizures that may be treatment-resistant. It is not yet well-understood whether outcome severity can be predicted by a variant's biophysical impact[3,7,14]. Biophysical studies of a small number of variants suggest that variants associated with early-infantile DEE have greater GoF than variants associated with S(F)NIS[7,8,15]. However, a small number of recurrent variants have been reported in both S(F)NIS and early-infantile DEE, indicating that the *SCN2A* variant itself is not the sole determinant of clinical severity[3,13,16,17].

Precision medicines are on the horizon for *SCN2A*-related disorders. Confident delineation of the biophysical impact and the phenotypes associated with GoF and LoF variants can ensure that a treatment aimed at reducing Na$_v$1.2 function (designed to address GoF) is not given to an individual with LoF, where it may cause significant deterioration. Further, accurate predictors of outcome severity are required to ensure that novel therapies are used only where existing treatments are likely to fail.

Here we review the phenotypic features of individuals with recurrent pathogenic *SCN2A* variants and study the biophysical profiles of selected variants using voltage clamp and DAPC. Our data show that DAPC is better than voltage clamp at predicting the biophysical impact of *SCN2A* variants. We confirm strong correlations between genotype, biophysical impact, and phenotypic group, but show that neither genotype nor biophysical impact can reliably predict outcome severity.

## Results

**Individuals and variants**. We evaluated phenotypic data of 179 individuals with 38 recurrent variants (Supplementary Data 1, Supplementary Table 1). The age range was 19 days–40 years (median 6 years). Current age was unknown in 87 individuals, although 42 were adults and four were older than age 2 years; for the remaining 41, there were sufficient data about age of seizure onset, epilepsy syndrome and development to classify phenotypic group.

Variants were de novo in 76 (42%; 2 mosaic in proband) and inherited in 73 (41%); inheritance was unknown in 30 individuals (17%). Six variants arose in both de novo and inherited forms (A263V, K908E, R1319Q, E1321K, V1325I, Q1531K). Eleven recurrent variants were chosen for biophysical analysis, spanning both early-infantile and later-onset phenotypes, and seen in individuals with a range of severities. To further understand the relationships between electrophysiological findings and severe phenotypes, we also studied biophysically two non-recurrent variants, K905N and R1629L, associated with severe early-infantile phenotypes. These two individuals are not included in the analysis of phenotypic data.

**Phenotypic group and correlation with genotype**. Using an age of seizure onset criterion to determine phenotypic group, 108 individuals had an early-infantile phenotype (seizure onset <age 3 months), 65 a later-onset phenotype (seizure onset ≥age 3 months) and 6 had ID/ASD. Ten of the 38 recurrent variants were identified in individuals with both early-infantile and later-onset phenotypes, reflecting an incomplete correlation between genotype and phenotypic group using this criterion alone.

In the second analysis, the criteria for an early-infantile phenotype were extended to include those with seizure onset between 3 and 24 months of age, with normal/near normal development prior to seizure onset and no epileptic spasms, features which can be seen in S(F)NIS. This led 36 individuals to be reclassified from the later-onset to the early-infantile phenotypic group. Only four of these individuals had tried SCB ASMs, none having seizure exacerbation, further supporting their early-infantile phenotypic classification. An early-infantile phenotype was therefore present in 144 individuals, later-onset in 29 and ID/ASD in 6 (Table 1). The genotype–phenotypic group correlation was stronger using these additional criteria, with 36/38 variants solely associated with one phenotypic group (30/38 early-infantile phenotype (Table 1 and Supplementary Table 1), 5/38 later-onset phenotypes, 1/38 ID/ASD without epilepsy). Two variants (R1435* and K503fs*) were each seen in one individual with a later-onset phenotype and one with ID/ASD without epilepsy. Given the improved correlation between genotype and phenotypic group, the remainder of this manuscript uses these extended criteria for phenotypic group allocation. The two non-recurrent variants were both associated with an early-infantile phenotype.

**Outcome severity and correlation with genotype**. Outcomes of individuals with early-infantile phenotypes were severe ($n = 35$), intermediate ($n = 23$), and benign ($n = 81$), as defined by criteria described in the "Methods" section and summarized in Supplementary Table 2; there were five unaffected individuals within affected families. All families with unaffected individuals carrying the variant had *SCN2A* variants that were not present in the Genome Aggregation Database (gnomAD). Seven individuals

**Table 1 Early-infantile and later-onset phenotypes of recurrent Na$_v$1.2 variants and their predicted functional impact.**

| Variant (N) | Un-affected (within early-infantile family) | Early-infantile benign | Early-infantile inter-mediate | Early-infantile severe | Later-onset (WS/un-classified infant-onset DEE) | Later-onset (LGS/un-classified childhood-onset DEE) | Later-onset (other epilepsy) | ID/ASD without epilepsy | DAPC prediction |
|---|---|---|---|---|---|---|---|---|---|
| Q153lK (8) | | + | | | | | | | GoF |
| L1563V (6) | | + | | | | | | | GoF |
| E1321K (5) | | + | | | | | | | GoF |
| Y1589C (9) | + | + | | | | | | | NT (pr. GoF) |
| M252V (4) | + | + | | | | | | | NT (pr. GoF) |
| R223Q (17) | | + | | | | | | | NT (pr. GoF) |
| L1330F (7) | | + | | | | | | | NT (pr. GoF) |
| V208E (5) | | + | | | | | | | NT (pr. GoF) |
| R36G (3) | | + | | | | | | | NT (pr. GoF) |
| R1882G (2) | | + | + | | | | | | NT (pr. GoF) |
| D343G (2) | | + | + | | | | | | NT (pr. GoF) |
| V261L (2) | | | + | | | | | | GoF |
| F1651C (2) | | | + | | | | | | NT (pr. GoF) |
| R1319Q (16) | + | + | + | + | | | | | GoF |
| A263V (11) | + | + | + | + | | | | | GoF |
| Q383E (3) | | + | + | + | | | | | GoF |
| V1325I (3) | | + | | + | | | | | GoF |
| K908E (4) | | + | | + | | | | | NT (pr. GoF) |
| V261M (3) | | + | | + | | | | | NT (pr. GoF) |
| S987I (2) | | + | | + | | | | | NT (pr. GoF) |
| R1629H (4) | | + | + | + | | | | | NT (pr. GoF) |
| R1882Q (8) | | | + | + | | | | | GoF |
| M1338T (2) | | | + | + | | | | | NT (pr. GoF) |
| E999K (5) | | | | + | | | | | GoF |
| R856Q (2) | | | | + | | | | | GoF |
| V423L (2) | | | | + | | | | | NT (pr. GoF) |
| S1336Y (2) | | | | + | | | | | NT (pr. GoF) |
| R1626Q (2) | | | | + | | | | | NT (pr. GoF) |
| G882E (2) | | | | + | | | | | NT (pr. GoF) |
| N212D (2) | | | | + | | | | | NT (pr. GoF) |
| E1211K (5) | | | | | + | | | | LoF |
| D195G (2) | | | | | + | | | | LoF |
| L1342P (5) | | | | | + | | | | NT (pr. LoF) |
| R220Q (2) | | | | | + | | | | NT (pr. LoF) |
| R853Q (14) | | | | | + | + | | | LoF |
| R1435* (2) | | | | | | | + | + | NT (pr. LoF) |
| K503fs* (2) | | | | | | | + | + | NT (pr. LoF) |
| R937C (4) | | | | | | | | + | NT (pr. LoF) |

Phenotypic groups allocated according to age of seizure onset (< or ≥ age 3 months), and additional clinical features if seizure onset (early-infantile if seizure onset was at ≥ age 3 months), and additional clinical features if seizure onset between age 3 and 24 months and normal/near normal development prior to seizure onset and no epileptic spasms, later-onset for all others with seizure onset ≥ age 3 months). The correlation between variant, phenotypic group and biophysical impact was less robust if phenotypic groups were defined only by age of seizure onset. Plus sign patient(s) assigned to early-infantile or later-onset phenotypic groups.
*Abbreviations:* DEE developmental and epileptic encephalopathy, ID/ASD intellectual disability/autism spectrum disorder without seizures, DAPC dynamic action potential clamp, GoF gain-of-function, LoF loss-of-function, pr. presumed, NT not tested, N number of individuals with the recurrent variant, WS West syndrome, LGS Lennox-Gastaut Syndrome.

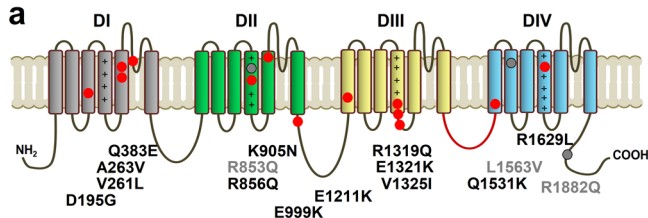

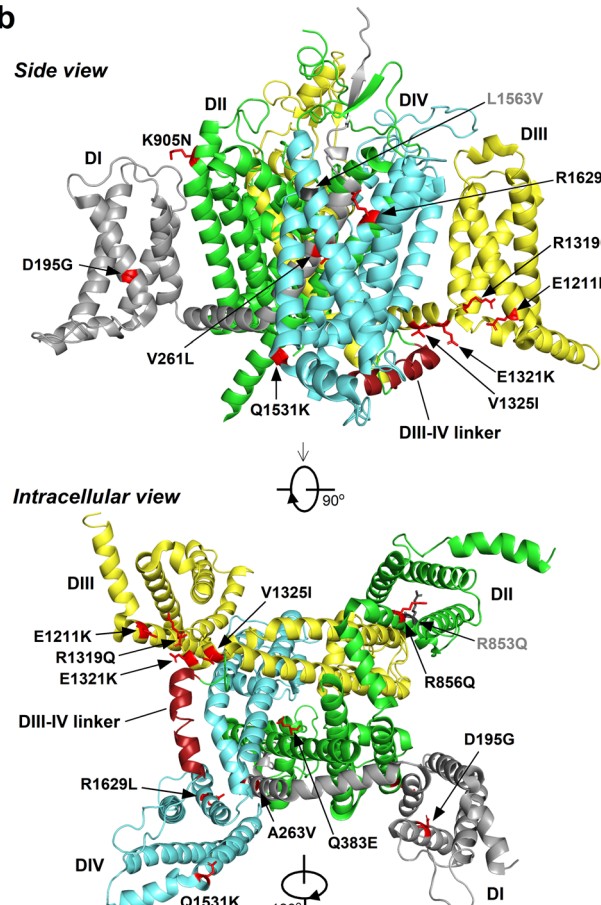

**Fig. 1 Nav1.2 channel and the locations of the functionally studied mutations.** Note the color-coded four domains (DI–DIV), the positive charges of segment 4 (S4) on DI–DIV, and the inactivation gate (DIII–DIV linker) marked in red. **a** Linear representation of the full-length Nav1.2 channel with the mutations studied here (red) or previously (gray)[7]. **b** Cryo-electron microscopy structure of Nav1.2[24]. The mutations are distributed in the pore module, the four peripheral voltage-sensing domains, the intracellular loops, and the C-terminus. Residues affected by the mutations are shown as sticks.

severe phenotypes. Variable severity within a single family was seen for three variants (A263V, Q383E, V1325I).

Later-onset phenotypes were identified in 29 individuals with seven different variants, and six individuals with three different variants had ID/ASD without epilepsy at last review. For five variants (D195G, R220Q, R853Q, E1211K, L1342P), all individuals ($n = 27$) had a later-onset phenotype, with broadly similar clinical features and severity within and between each of these variants. Seizure onset was between 3 months and 3 years (median 8 months). Epilepsy syndromes at presentation were West Syndrome (14), unclassified DEE (6), Lennox–Gastaut syndrome (1) or unknown (6). Seven individuals had seizure exacerbations on SCB ASMs, but five reported some benefit. Two variants, R1435* and K503fs* were associated with both a later-onset phenotype and ID/ASD without epilepsy, each having one individual in each phenotypic group. The two children with epilepsy had delayed development prior to onset of tonic-clonic seizures (epilepsy syndrome not reported) at age 2 years 11 months and age 6 years. The R937C variant was seen only in individuals with ID/ASD without seizures; all four individuals had developmental delay/ID and two had ASD.

**Biophysical characterization of Nav1.2 channel variants using voltage clamp recordings.** The 13 variants studied (Fig. 1) included 11 associated with an early-infantile phenotype and two with a later-onset phenotype (E1211K, D195G). The early-infantile variants included four consistently associated with a severe outcome (early-infantile consistently severe: R1629L, E999K, R856Q, and K905N), two consistently associated with a benign outcome (early-infantile consistently benign: E1321K and Q1531K), and five associated with outcomes of variable severity (early-infantile variable severity: A263V, V1325I, V261L, R1319Q, and Q383E). The variants are localized in channel regions associated with specific functions, including voltage sensing: R856Q, R1319Q, and R1629L (in segment 4 of domain II (S4$_{DII}$), S4$_{DIII}$, and S4$_{DIV}$, respectively); channel gating: E1211K, Q1531K, and D195G (in S1$_{DIII}$, S1$_{DIV}$, and S3$_{DI}$, respectively); fast inactivation: E1321K and V1325I (in the S4–S5$_{DIII}$ linker specifically implicated in forming the inactivation gate receptor[18]); pore module[19]: A263V, V261L (both in S5$_{DI}$), K905N (S5$_{DII}$), and Q383E (S5–S6DI turret loop:); and protein trafficking to the axon initial segment (AIS)[20]: E999K (DII–DIII linker) (Fig. 1).

In voltage clamp experiments, we determined the biophysical characteristics of the variants relative to wild-type. The sodium current density values of the variants, providing the number of channels relative to membrane surface area, were unchanged, indicating that the mutations did not alter channel expression (Supplementary Fig. 1 and Supplementary Table 3). The investigation of the activation and steady-state inactivation demonstrated that the mutations result in mixed effects on gating in the early-infantile and later-onset groups (Fig. 2, Supplementary Fig. 2, Supplementary Tables 3, 8). Like previous studies[7,21], we found that enhancement of the persistent Nav1.2 currents relative to wild-type, known to contribute to channel GoF, is solely associated with early-infantile phenotypes (Supplementary Fig. 3, Supplementary Tables 3, 8). In the voltage range between −40 and +25 mV, the time course of fast inactivation was slower for the R1629L, V1325I, and Q1531K variants compared to wild-type, consistent with a GoF effect (Supplementary Fig. 4 and Supplementary Table 3). Recoveries from inactivation, assessed with a two-pulse protocol, were fast or slow in the early-infantile groups, and slow in the later-onset group, relative to the wild-type channel (Supplementary Fig. 5, Supplementary Tables 3, 8).

were deceased, all severely affected prior to death. Three individuals with seizure onset at/after age 3 months did not have a normal developmental outcome (intermediate phenotype). The *SCN2A* variant did not consistently predict outcome; clinical severity of the early-infantile phenotype was consistent between individuals for only 18/30 (60%) recurrent variants (Table 1). Six variants were consistently associated with a severe outcome, including one (R856Q) for which both individuals were deceased, and nine consistently associated with a benign course. Variation in severity was noted for the remaining 12/30 (40%) variants, seven of which were seen in individuals with both benign and

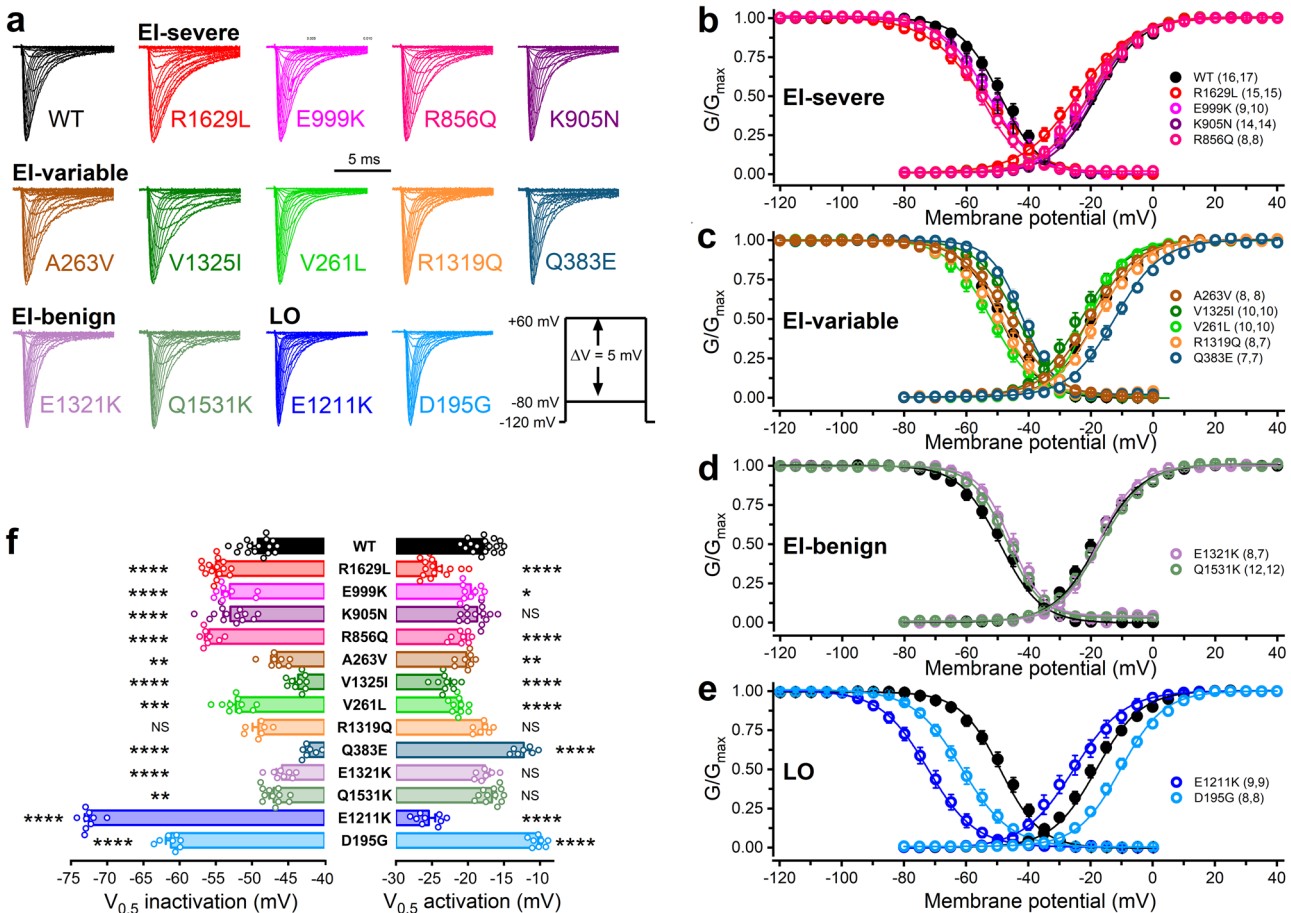

**Fig. 2 Voltage dependence of activation and inactivation of wild-type (WT) and mutant Na$_v$1.2 channel variants associated with early-infantile (EI) severe/variable/benign, or later-onset (LO) *SCN2A* epilepsy. a** Representative Na$_v$1.2 current traces. Note that only the first 10 ms of the current traces elicited in the voltage range between −80 and +20 mV are shown. The voltage protocol (inset) consisted of 40-ms depolarizing voltage steps, elicited at 1 Hz in 5-mV increments, in the voltage range between −80 and +60 mV, from a holding potential (HP) of −120 mV. Current traces of individual variants were normalized to the same amplitude. Horizontal scale bar: 5 ms. **b**–**e** Activation and inactivation curves were obtained by non-linear least-squares fits of Boltzmann equations (Eq. (1)) to normalized conductance ($G/G_{max}$) data of wild-type Na$_v$1.2 (black solid circle) and mutant Na$_v$1.2 channels (colored open circles). See the parameters of the fits in Supplementary Table 3; *n* values, the number of independent experiments for inactivation and activation, respectively, are shown in parentheses. **f** Mean half-maximal voltages of activation and inactivation ($V_{0.5,act}$ and $V_{0.5,inact}$, respectively) of wild-type Na$_v$1.2 (black bar) and mutant Na$_v$1.2 channels (colored open bars). Data for individual cells are shown in open circles. Relative to wild-type, hyperpolarising shifts of the $V_{0.5,act}$ values are consistent with increased channel opening and gain-of-function (GoF), whereas depolarizing shifts are consistent with reduced channel opening and loss-of-function (LoF); hyperpolarising shifts of the $V_{0.5,inact}$ values are consistent with reduced channel availability (LoF), whereas depolarizing shifts are consistent with increased channel availability (GoF). Data shown are mean ± SEM; *n* values for all variants are shown in **b**–**e**. *$P < 0.05$, one-way ANOVA, followed by Dunnett's post-hoc test; the detailed statistical evaluation of biophysical characteristics of the variants is shown in Supplementary Table 3.

Overall, voltage clamp data predicted GoF for five variants (A263V, V1325I, R1319Q, E1321K), LoF for D195G, and no change for K905N. Almost half of the variants (R1629L, E999K, R856Q, V261L, Q383E, and E1211K) exhibited mixed GoF and LoF biophysical characteristics, obstructing the process of reliably predicting the overall impact of the mutation from voltage clamp data alone.

We implemented two previously published tools, which have utilized voltage clamp data for prediction of biophysical impact or severity of *SCN2A* variants. The first, a machine learning-based statistical model for function prediction[22], providing GoF or LoF probability for the individual Na$_v$1.2 variants studied here and previously[7]. This method resulted in unreliable functional predictions because 12 of the 16 variants were previously included in the training dataset of the model[22] (Supplementary Table 8). In the second, we calculated the clinico-electrophysiological severity score of Na$_v$1.2 variants (CESSNa$^+$ score) to determine the extent

of changes of selected biophysical properties relative to wild-type, according to a previously published method[15] (Supplementary Table 4). Mean CESSNa$^+$ scores of the early-infantile-severe, early-infantile-variable, and early-infantile-benign groups were statistically not significantly different ($P = 0.46$; one-way ANOVA).

**Predicting the effects of SCN2A variants on action potential firing using DAPC approach.** We assessed the action potential firing activity of the hybrid AIS model neuron incorporating virtual conductances and wild-type or mutant gNa$_v$1.2 expressed in a Chinese Hamster Ovarian (CHO) cell (Fig. 3, Supplementary Fig. 6, Supplementary Tables 5, 7). The stimulus current-firing frequency (input-output) relationships demonstrate that all the early-infantile variants, except the K905N, which showed no change, result in a significantly higher action potential firing than wild-type (Fig. 3b). Firing activities and input-output relationships

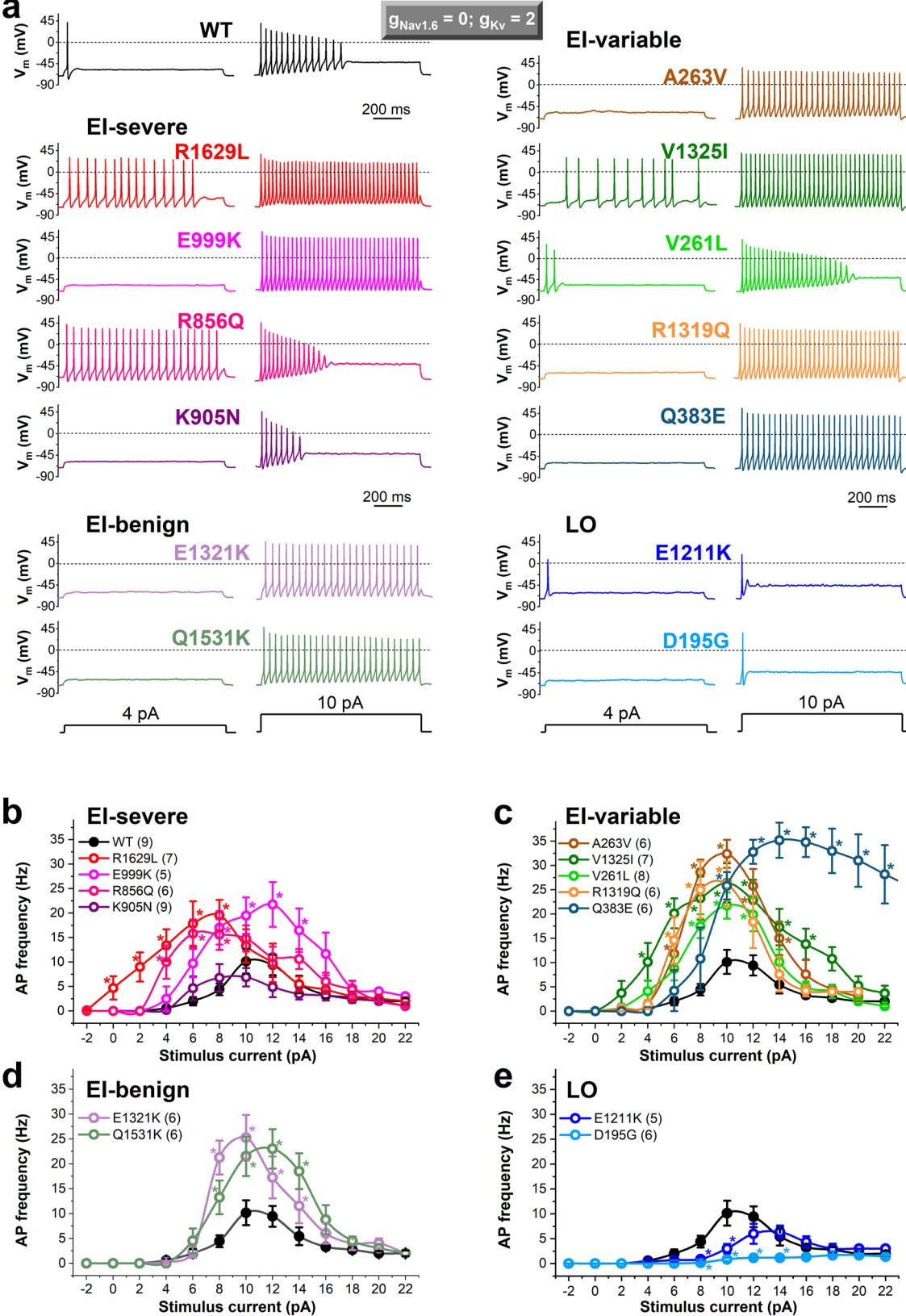

associated with early-infantile consistently benign, variable severity, and consistently severe mutant channels were all very similar to each other. The later-onset phenotype variants had reduced firing and showed increased rheobase compared to wild-type (Fig. 3b). We assessed the contribution of Na$_v$1.2 variants to the action potential morphology by determining the rheobase,

threshold, upstroke velocity, amplitude, width, and decay time. Action potential characteristics of several variants were altered relative to wild-type (Supplementary Table 6); the increased width and decay time of E999K, R856Q, A263V, V1325I, Q383, and Q1531K was correlated with increased persistent Na$_v$1.2 current for these variants in voltage clamp mode (Supplementary Fig. 3).

**Fig. 3 Dynamic action potential clamp experiments implementing heterologously expressed wild-type (WT) or mutant Na$_v$1.2 channels associated with early-infantile (EI) or later-onset (LO) *SCN2A* epilepsy.** In the axon initial segment (AIS) compartment model, the Na$_v$1.6 conductance (gNa$_v$1.6) was set to zero, whereas the potassium channel conductance (gK$_v$) was adjusted to 200% (gNa$_v$1.6 = 0, gK$_v$ = 2, respectively), enabling a more efficient action potential (AP) repolarisation during firing compared to a gK$_v$ setting of 1 (Supplementary Fig. 6). **a** Representative examples of AP firing in response to 4 and 10 pA step current stimuli. Note that each Na$_v$1.2 variant resulted in a unique firing profile relative to WT. **b–e** Input–output relationships demonstrating the effect of increasing stimulus strength on action potential frequency in the presence of Na$_v$1.2 channel variants (WT, black solid circle; mutants, colored open circles) associated with early-infantile (EI)-severe (**b**), EI-variable (**c**), EI-benign (**d**), or later-onset (LO) (**e**) *SCN2A* epilepsy. Firing was elicited by stimulus current steps in the range between −2 and 22 pA, in 2 pA increments. Data shown are mean ± SEM; *n*, the number of independent experiments between parentheses. Firing frequencies relative to WT were assessed using two-way ANOVA followed by Dunnett's post-hoc test; asterisks indicate *P* < 0.05 (for individual *P* values see Supplementary Table 5). Note the decreased or increased rheobase with the EI-severe R1629L and later-onset D195G variants, respectively. The statistical evaluation of the action potential morphology is shown in Supplementary Table 6.

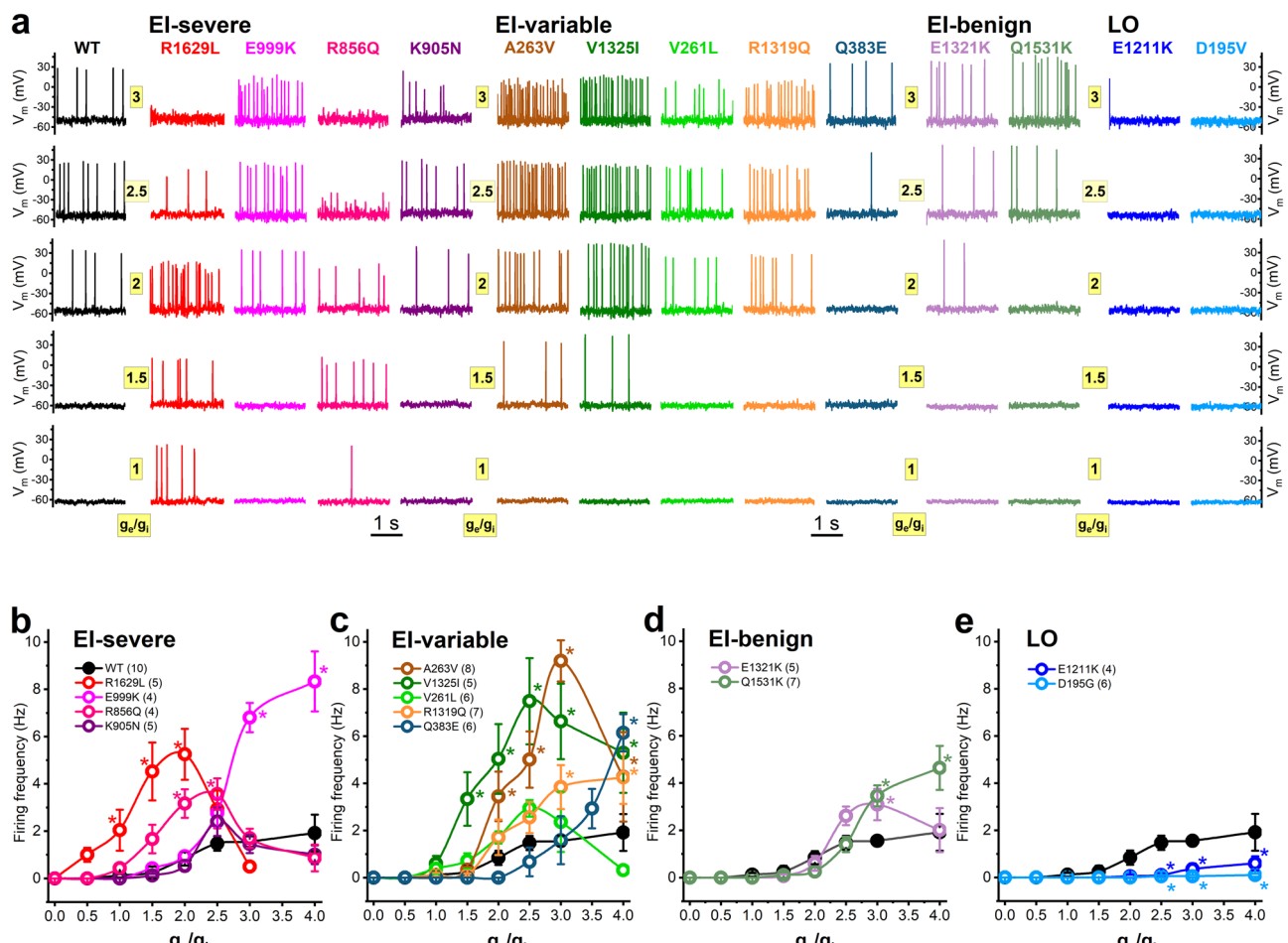

**Fig. 4 Action potential firing of the axon initial segment hybrid neuron incorporating Na$_v$1.2 variants in response to synaptic conductance input in DAPC experiments. a** Representative action potential firing elicited with $g_e/g_i$ ratios of 1, 1.5, 2, 2.5, and 3 in the early-infantile (EI)-severe, EI-variable, EI-benign and later-onset (LO) groups. **b–e** Input–output relationships in the presence of Na$_v$1.2 channel variants (WT, black solid circle; mutants, colored open circles) associated with EI-severe (**b**), EI-variable (**c**), EI-benign (**d**), or later-onset (LO) (**e**) *SCN2A* epilepsy. Two-way ANOVA, followed by Dunnett's post-hoc test, was used to compare the AP firing frequencies elicited by scaled excitatory to inhibitory conductance ratios ($\mathbf{g}_e/\mathbf{g}_i$) in the presence of Nav1.2 variants; asterisks indicate *P* < 0.05 (see individual *P* values in Supplementary Table 7). Note the increased firing activity in the early-infantile severe/variable groups compared to wild-type, whereas later-onset variants result in an almost complete loss of firing. Data shown are mean ± SEM; *n*, the number of independent experiments between parentheses.

We also tested the impacts of Na$_v$1.2 variants on hybrid neuron firing using scaled ratios of excitatory and inhibitory synaptic conductances ($g_e/g_i$), as this is a more biologically realistic synaptic current stimulus than the step stimulus trialed above[23]. Representative action potential firing activities elicited with $g_e/g_i$ ratio values between 0.5 and 3 and a virtual conductance setting of $gNa_v1.6 = 0/gK_v = 1$ are shown in Fig. 4. The input–output relationships (Fig. 4b), were similar to those elicited by step

stimuli (Fig. 3b), further validating the unique firing profiles due to distinct Na$_v$1.2 mutations.

Taken together, in DAPC mode, all early-infantile phenotype variants except K905N resulted in GoF and both later-onset phenotype variants produced LoF. Thus, DAPC reliably predicted phenotypic group for all but one variant. However, it could not differentiate benign from severe early-infantile phenotypes.

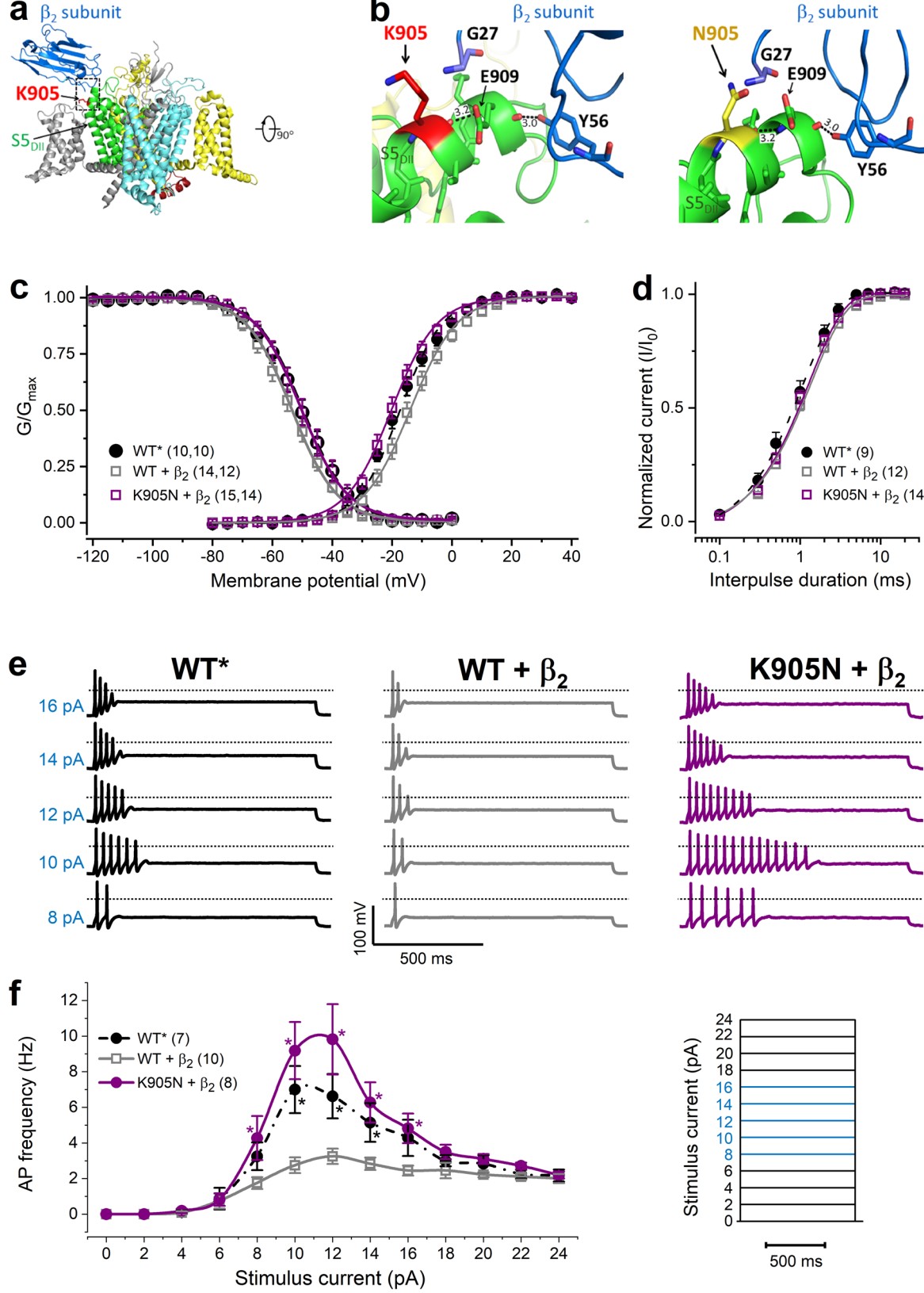

**3D structural modeling.** To better understand the molecular basis for the functional data from voltage clamp and DAPC assays, we mapped the variants onto a 3D model of the Na$_v$1.2 channel[24]. The detailed views of the channel, the predicted structural changes due to the mutated residues, and the interpretations of the effects caused by these modifications are detailed in Supplementary Fig. 7. Briefly, the D195G mutation disrupts polar interactions in S3$_{DI}$; V261L and A263V mutations affect hydrophobic interactions in DI; Q383E affects the key E384 residue in the DEKA selectivity filter[16]; R856Q, R1319Q, and R1629L affect gating-charge carrying R residues, which are key for voltage sensor movements; E1211K involves the change of a

**Fig. 5 Gain-of-function characteristics of the K905N variant are revealed in the presence of β2 subunit. a** 3D structure of the wild-type $Na_v1.2$ channel-β2 subunit complex (PDB accession no. 6J8E[24]) showing the four main channel domains DI–IV (DI, gray; DII green; DIII, yellow; DIV, cyan) and the β2 subunit (blue). **b** Structure-predicted interactions between the α1-subunit $S5_{DII}$ region and the β2-subunit. Expanded views of the helix in segment 5 of domain II ($S5_{DII}$) (boxed area in A) showing the positively charged K905 (wild-type, left) and the polar, uncharged side chain residue N905 (mutant, right). All *residues* within 5 Å distance from K905 or N905 are shown in stick representation (*blue*: nitrogen, *red*: oxygen). **c** Activation and inactivation curves, obtained by non-linear least-squares fits of Boltzmann equations (Eq. (1)) to data points (WT alone, black solid circle; WT + β2, gray open square; K905N + β2, purple solid circle); see the parameters of the fits in the Supplementary Table 9; $n$ values, the number of independent experiments for inactivation and activation (first and second number, respectively), are shown in parenthesis. **d** Recovery from fast inactivation. The time constants of recovery ($\tau$), included in Supplementary Table 9, were obtained by fitting a single exponential function to the data (Eq. (3)) (symbol definition same as in **c**). Statistical evaluation of data in **c** and **d** (one-way ANOVA followed by Dunnett's post-hoc test) is shown in Supplementary Table 9. **e** Representative examples of action potential firing in response to depolarizing step current stimuli (8–16, 2 pA steps, 1 s duration) in DAPC experiments using $gNa_v1.6 = 0$ and $gK_v = 2$ settings; the corresponding step stimuli are highlighted in blue in the stimulus protocol shown in (**f**); the dotted lines indicate 0 mV. **f** Input–output relationships showing the dependence of the action potential firing on stimulus current magnitude. Firing of the hybrid AIS neuron incorporating WT (black solid circle), WT + β2 (gray open square), or K905N + β2 (purple solid circle) was elicited by current steps in the range between 0 and 24 pA, in 2 pA increments (inset protocol). Data in **c–f** are mean ± SEM; $n$ values, the number of independent experiments, are shown in parentheses. Differences in firing activity elicited at various current amplitudes were evaluated with two-way ANOVA followed by Dunnett's post-hoc test; asterisks indicate $P < 0.05$ (for individual $P$ values see Supplementary Table 10). Action potential characteristics are summarized in Supplementary Table 11.

highly conserved negative residue to positive in $S4_{DIII}$; E1321K and V1325I affect coupling interactions between the voltage sensor and the pore[25], and residues involved in fast inactivation[19] in S4–$5_{DIII}$; and Q1531K results in the change of a conserved residue with polar uncharged side chain to a positive residue in $S1_{DIV}$. The structure of the DII–DIII linker carrying the E999K mutation is currently not resolved.

The functional impact of K905N, a variant associated with early-infantile *SCN2A* epilepsy, could not be resolved by either voltage clamp or DAPC. Assessment of the 3D structure of the wild-type Nav1.2 α1 subunit in complex with the β2 subunit suggests that the helix in segment 5 of domain II ($S5_{DII}$) and β2 establish a limited number of specific contacts, including a hydrogen bond between the side group of the Tyr56 in β2 and the carbonyl-oxygen of Glu909 in $S5_{DII}$ (Fig. 5a, b); furthermore a disulfide bond between C910 on DII S5 and C55 on β2 (not shown); and polar interactions between D917, 918 on DII S5 and R135 on β2 (not shown)[24]. The positively charged K905 residue is localized in $S5_{DII}$ and is directed outward from the four-domain arrangement (Fig. 5a).

**Predicting the functional impact of K905N variant using co-expression of α1 and β2 subunits.** We hypothesized that the K905N mutation indirectly destabilized key electrostatic interactions in $S5_{DII}$ and/or between $S5_{DII}$ and the β2 subunit (Fig. 5b). To test the impact of β2 on Nav1.2 channel variant function, we co-expressed the wild-type or the K905N α1 pore-forming subunit and the β2 subunit (WT + β2 and K905N + β$_\beta$, respectively) in CHO cells. Simultaneously, we also repeated the experiments with CHO cells transfected with wild-type α1 subunit alone (WT*) to enhance experimental control. In voltage clamp experiments, the activation, inactivation, and recovery from inactivation parameters for WT* channels (Supplementary Table 9) were indistinguishable from those of the wild-type channels shown in Fig. 2, Supplementary Fig. 5, and Supplementary Table 3. Co-expression of the β2 subunit did not affect current density (Supplementary Table 9) but shifted the voltage dependence of the wild-type and K905N variants (Supplementary Table 9). The depolarizing shift of the WT + β2 activation curve (Fig. 5c) agrees with published data[26]. Relative to WT + β2 channels, the activation and inactivation curves of K905N + β2 exhibited hyperpolarizing and depolarizing shifts, respectively, which correspond to gain-of-function (Fig. 5c), whereas the time course of recovery from fast inactivation for K905N + β2 was unchanged (Fig. 5d and Supplementary Table 9). In DAPC mode, the hybrid cell model incorporating K905N + β2 or WT* current

achieved significantly higher firing frequencies over a range of step stimuli relative to WT + β2 (Fig. 5e and Supplementary Table 10). Action potential characteristics were similar except the decreased threshold of K905N + β2 relative to WT + β2 (Supplementary Table 11).

Our data suggest that the decreased excitability of the hybrid neuron incorporating WT + β2 current relative to WT* is due to the interaction between the heterologously expressed α1 and β2 subunits; the K905N mutation hinders these interactions, resulting in GoF.

## Discussion

This clinical and functional characterization of recurrent *SCN2A* variants addresses prerequisite physiological insights for targeted therapeutics: how do we determine biophysical impact and predict outcome severity? Candidacy for clinical trials will be based on whether the individual's *SCN2A* variant produces GoF or LoF, and whether outcomes are likely to be poor despite existing treatments. We confirm that genotype, phenotypic group and DAPC are accurate predictors of the biophysical impact of the *SCN2A* mutation on the Nav1.2 channel, and that DAPC overcomes limitations of the widely used voltage clamp technique in unambiguously determining GoF or LoF. However, neither genotype nor biophysical impact predict outcome severity in the early-infantile phenotypes.

Our study confirms those of previous studies of a strong association between *SCN2A* phenotypic group and biophysical impact on the Nav1.2 channel. Here we also address two concerns about the robustness of this association, improving its reliability.

The first is whether there is an experimental approach that can provide more reliable predictions of the overall functional impact of an *SCN2A* mutation than the most widely used method, voltage clamp. There is evidence that voltage clamp can be inefficient for predicting the overall functional impact of the variants in instances where the channel exhibits mixed biophysical characteristics that could result in either GoF or LoF[7,9]. In this study, voltage clamp was unable to confidently resolve the biophysical impact of almost half the variants studied. In contrast, the DAPC assay provided confident predictions of GoF or LoF for all but one variant, which were consistent with the expected clinical phenotypes (Supplementary Table 8), strengthening the findings of our previous work[3,7]. Although detailed voltage clamp data are still important in some instances, DAPC is a better choice for a rapid and unambiguous assessment of novel *SCN2A* variants identified in potential candidates for clinical trials, where an

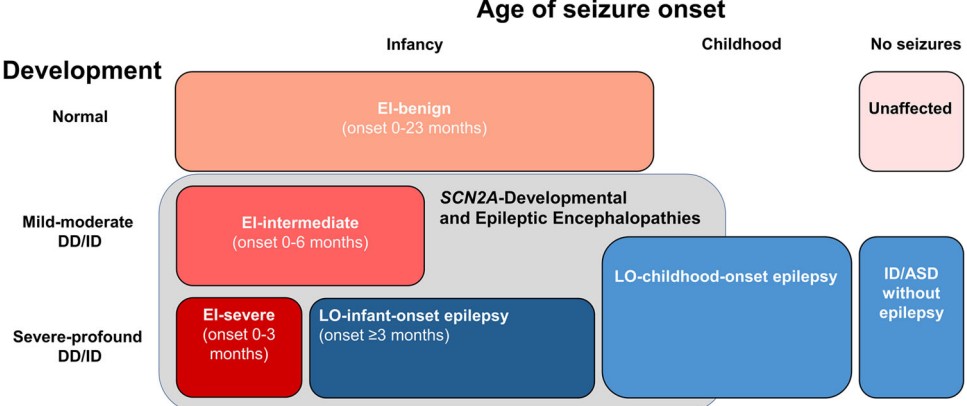

**Fig. 6 The phenotypic spectrum of SCN2A-related disorders.** Pink/red boxes indicate phenotypes associated with gain-of-function (GoF) of the Nav1.2 channel, and blue boxes loss-of-function (LoF) phenotypes; Onset refers to age of seizure onset. The phenotypic groups associated with more severe epilepsies are sometimes grouped as *SCN2A* developmental and epileptic encephalopathies (gray box), which encompasses both GoF and LoF phenotypes. The early-infantile (EI) phenotypes can be considered a spectrum of clinical severity associated with biophysical GoF. While the later-onset (LO) epilepsy phenotypes and ID/ASD without epilepsy are both associated with LoF, it is not clear whether these represent a spectrum of severity (related to partial or total loss of channel function), or are distinct disorders caused by different mechanisms of channel hypofunction (for example pure LoF vs dominant negative effect).

efficient and cost-effective evaluation of the variant's impact is desired.

The functional impact of one variant, K905N, was not resolved by either voltage clamp or DAPC. Previous studies have demonstrated that the $\beta_2$ subunit exerts depolarizing effects on Na$_v$1.2 gating and alters cell surface expression of Na$_v$1.2[26,27], and our molecular modeling suggested the K905N variant may alter α1 and β2 subunit interactions. We therefore hypothesized that co-expression of the α1 and β2 subunit interactions would be needed to reveal the impact of this mutation; our results support this hypothesis, revealing a GoF, consistent with that expected for the phenotype. This finding highlights that molecular modeling and subunit co-expression may be useful tools in determining variant impact for the minority of cases in which this is not resolved by DAPC.

The second is determining the most reliable clinical criteria for distinguishing phenotypes associated with GoF or LoF biophysical impact. Although our and other studies show that seizure onset before age 3 months is seen only with GoF variants, seizure onset at/after age 3 months can be seen both with LoF and GoF variants (2,3,13,15,28–31, and this study). Thus, if age at seizure onset is used as the sole factor for classifying a phenotype as early-infantile or later-onset, the phenotypic groups does not reliably correlate with genotype or biophysical impact. As GoF variants with seizure onset at/after age 3 months are predominantly seen in those with benign outcomes (i.e., S(F)NIS), previous studies have separated S(F)NIS from DEE to improve the strength of the correlation of biophysical impact with age alone. However, this can be problematic in clinical practice, particularly at initial presentation. Not all individuals with GoF variants and seizure onset at/after age 3 months have benign outcomes (intermediate outcomes seen with seizure onset as late as 6 months). Further, assuming all presentations with seizure onset at/after age 3 months are due to LoF may mean SCB ASMs are avoided rather than used preferentially. Previous studies have shown that SCB ASMs may be needed for seizure control even in the benign phenotypes[3,16]. In this study, we show that the genotype–phenotype correlation is improved by using an additional criterion, such that the early-infantile phenotypic group was defined as including individuals with seizure onset prior to age 3 months, and individuals with seizure onset at or after age 3 months with normal/near normal development prior to seizure

onset and no epileptic spasms (Fig. 6). This definition means that the early-infantile group is more reliably considered analogous to GoF. However, given this phenotypic group now includes individuals with seizure onset beyond early infancy, it may be better named the GoF phenotypic group.

With current knowledge, using clinical features to infer biophysical impact (if functional data are not available) is probably sufficiently reliable for clinical practice, where prompt treatment decisions are required. However, we acknowledge there may be individuals outside this study for whom these criteria do not correctly predict biophysical impact. Given the potential for exacerbation of the condition if the wrong mechanism is assumed, biophysical confirmation of presumed GoF or LoF remains prudent in the clinical trial setting.

Our study included only a small number of recurrent variants associated with later-onset epilepsies and ID/ASD. These phenotypes are thought to arise because of LoF, as previously shown for some variants (e.g. R853Q), although conflicting results have been seen for other variants[7,8,10,32]. We show using DAPC that the E1211K variant, previously interpreted as showing either GoF or LoF[9], produced LoF, as expected given the associated later-onset phenotype. We were, however, unable to resolve whether the different LoF phenotypes are distinct entities or represent a single group with broad phenotypic variability. Several LoF recurrent variants (missense) have been seen only in children with severe epilepsies beginning in mid-late infancy, and others (missense or truncating) were seen either in individuals with childhood-onset epilepsies or ASD/ID, providing preliminary suggestions of a possible distinction in terms of mechanism and/or severity between those with LoF phenotypes who have seizures during infancy and those who do not. In this study, we did not perform biophysical analyses of variants seen in individuals with childhood-onset epilepsy or ID/ASD to evaluate if variants causing these LoF phenotypes differ in the degree of Na$_v$1.2 function loss (e.g. partial versus complete) compared with LoF variants causing mid-late infancy epilepsies. The cellular basis of how reduced Na$_v$1.2 function leads to seizures is not well understood. In a Na$_v$1.2 deficient, heterozygous *Scn2a*$^{+/-}$ mouse model, downregulation of potassium channels contributes to the elevated intrinsic excitability of cortical and striatal neurons[33]. Complete knockout of *Scn2a* affects potassium channel repolarisation in prefrontal pyramidal cells of a conditional mouse

model. This highlights the subtle roles of $Na_v1.2$ and potassium channels in electrogenesis, and raises the possibility that secondary or compensatory mechanisms, rather than the differences in function of the $Na_v1.2$ channel itself, may explain differences in clinical features in the LoF phenotypes[34].

Previous work has suggested that biophysical impact of a variant may correlate with outcome severity in the early-infantile/ GoF phenotypic group but reports of variable severity for a small number of recurrent variants has raised concerns that the variant may not be a reliable predictor[15–17]. Here, we show significant variability in outcome severity between individuals with the same variant for 40% of the early-infantile variants studied, including seven variants in which both benign and severe outcomes were seen, and three with variable severity between members of the same family. This variability appears more common than previous reports have suggested, confirming that the variant itself is an unreliable predictor of outcome severity. Further, the biophysical impact of variants correlated poorly with phenotypic severity for early-infantile/GoF variants. Our previous evaluation of a smaller number of variants[7], and a study introducing voltage clamp data-derived clinico-electrophysiological severity (CESSNa[+]) scoring[15], indicated a possible relationship between the degree of electrophysiological dysfunction and phenotypic severity. However, when considering a larger number of variants and individuals with each variant, we found that neither DAPC, nor voltage clamp including the CESSNa[+] score could reliably segregate early-infantile variants by severity. While this is not surprising for variants associated with outcomes of variable severity, both techniques also failed to differentiate variants associated with consistently benign or consistently severe outcomes.

Factors influencing severity are not clear and were not studied here. The role of modifying genes, epigenetic factors, variability in *SCN2A* allelic expression, and the impact of existing ASM all warrant further consideration. The strength of association between early clinical features and outcome severity should also be examined, as these may reveal robust early predictors of severity, which will be critical to enable prompt administration of novel therapies in clinical trials to individuals whose outcomes will be poor with existing treatments.

## Methods

**Standard Protocol approvals and patient consents**. The study was approved by the Human Research Ethics Committees of the Royal Children's Hospital and Austin Health Melbourne, University Medical Center Groningen, State Medical Association of Berlin, and the Simons Foundation. Written informed consent was obtained for all individuals whose previously unpublished clinical data is presented here.

**Identification of variants and patients**. The medical literature, *SCN2A* International Natural History Study (NHS) database, Florey Institute's Ion Channels Laboratory database, and Simons Searchlight database (SSDb) (https://www.sfari.org/resource/simons-searchlight/) were searched to identify all recurrent *SCN2A* variants with clinical information available on affected individuals. Individuals from the SSDb were only included if the individual had not been previously reported and was not in the NHS, to avoid duplicate reporting.

Forty-one recurrent *SCN2A* variants were identified. Three variants were excluded from further study. One variant (R187W), identified in two individuals, was excluded because no clinical information was available for one (father of proband), and the parents of the proband (who had febrile seizures plus) both had febrile seizures, raising the possibility of additional contributing factors for the proband's phenotype[35], and for two variants (R379H and C959*), there was no available clinical data for any individual with that variant (two individuals for each variant). For three additional variants (R1629H, R1626Q, R853Q), two individuals with each variant were excluded due to insufficient clinical information.

Thirty-eight recurrent *SCN2A* variants were included with clinical information available on at least two individuals per variant. Eleven recurrent variants were selected for biophysical analysis and 3D structural modeling, spanning both early-infantile and later-onset *SCN2A*-related epilepsy phenotypes. Seven variants have not been previously studied biophysically, whereas the remaining four (A263V, E999K, E1211K, and R1319Q) have been studied with voltage clamp (Supplementary Data 2). To further understand the relationships between

electrophysiological *findings* and *severe* phenotypes, we also analyzed two non-recurrent variants (K905N, R1629L), each identified in a single individual with early-infantile-DEE[1,36].

**Clinical data**. Clinical information for 18 new, and updated data on 3 reported, individuals was obtained from interviews with parents and review of home videos by a pediatric epileptologist (KBH), and medical records. Clinical data for the remaining patients were obtained from the literature ($N = 159$) or Simons Foundation database ($N = 1$).

Patients were allocated to one of three phenotypic groups; early-infantile, later-onset or ID/ASD (without epilepsy) (Supplementary Table 1), using the previously reported phenotypic spectra for each group[2,3,5,7].

Group allocation was done twice, using different criteria. In the first allocation, age of seizure onset was used as the major criterion. Early-infantile was considered to be prior to age three months, later-onset at or after three months, and ID/ASD as having no seizures, given the three months age cut-off has been previously reported as distinguishing these groups, at least for *SCN2A*-related DEEs[3]. If age of seizure onset was unknown, we considered individuals with early-infantile epileptic encephalopathy (EIEE), epilepsy of infancy with migrating focal seizures, or S(F) NIS (previously known as benign (familial) neonatal-infantile seizures (B(F)NIS)) to have an early-infantile phenotype, and individuals with West syndrome or Lennox-Gastaut syndrome at presentation, a later-onset phenotype. Unaffected individuals with *SCN2A* variants who had affected family members with S(F)NIS were considered to have early-infantile phenotypes, as incomplete penetrance/ variable expressivity is reported in S(F)NIS[37].

In the second allocation, we used additional criteria, designed to include all individuals with S(F)NIS in the early-infantile group given S(F)NIS may have seizure onset after age 3 months, although has a number of features in line with the phenotypes of those with earlier seizure onset, including focal seizures, reports of benefit with SCB ASMs, and biophysical studies showing GoF[2–5,29]. We therefore considered individuals to have an early-infantile phenotype if seizure onset was before age 3 months, or if the individual had seizures beginning at age 3–24 months without epileptic spasms, and normal development prior to seizure onset or unknown development prior but normal/near normal developmental outcome. All other individuals with epilepsy onset ≥ age 3 months were classified as having a later-onset phenotype.

The early-infantile phenotypic group was subdivided according to severity, based on developmental outcome, persistence of seizures and other neurologic impairments after infancy, into severe, intermediate, benign, and unaffected groups[1] (Supplementary Table 2). Individuals in the benign group had normal development and no seizures or other neurologic symptoms after age two years. Those in the severe group had severe developmental delay or ID, with or without ongoing seizures and other neurologic features such as abnormal tone or movement disorders after age 2 years. Individuals in the intermediate group had persistence or development of neurologic symptoms after age 2 years (epilepsy, episodic ataxia or other, as these are not consistent with S(F)NIS), and/or developmental impairment that was not severe (range normal-moderate impairment). The later-onset phenotypic group was subdivided where possible by epilepsy syndrome at presentation.

**Site-directed mutagenesis, cell culture, and transfection**. Mutations were introduced into the adult isoform of the $Na_v1.2$ complementary DNA[38] using QuikChange site-directed mutagenesis (Agilent Technologies, Santa Clara, CA) with custom made primers (Bioneer Pacific Australia) (see sequences in Supplementary Table 12). All mutant $Na_v1.2$ constructs were verified by automated DNA sequencing (Australian Genome Research Facility, Melbourne).

Chinese hamster ovarian (CHO) cells were cultured in T25 cm² flasks (BD Biosciences, San Jose, CA, USA) containing Dulbecco's Modified Eagle Medium: Nutrient Mixture F-12 (Thermo Fisher Scientific, Waltham, MA) supplemented with 10% (v/v) fetal bovine serum (Thermo Fisher Scientific) and 50 IU/ml penicillin (Thermo Fisher Scientific) in a 37 °C incubator with a humidified atmosphere of 5% $CO_2$ in air. At ~80% confluency, the cells were transiently co-transfected with wild-type (adult isoform)[38] or mutant $Na_v1.2$ channel (5 μg) and enhanced green fluorescent protein (eGFP; 1 μg; Clontech, Mountain View, CA) gene construct using Lipofectamine 3000 Reagent (Thermo Fisher Scientific), and incubated at 37 °C in 5% $CO_2$ for 24 h. Following TrypLE Express Reagent (Thermo Fisher Scientific) treatment, the cells were plated on 13 mm diameter glass coverslips (Menzel-Gläser, Thermo Fisher Scientific), and incubated at 30 °C in 5% $CO_2$. Co-transfections involving the $β_2$ subunit were performed using 4 μg wild-type or K905N sodium channel $α_1$ subunit, 3 μg human $β_2$ subunit (NCBI Reference Sequence: NM_004588.5; Origene Technologies, Rockville, MD) and 1 μg eGFP.

**Voltage clamp experiments**. Voltage clamp recordings were performed at room temperature (23 ± 0.5 °C) 48–72 h after transfection. The cells were super-fused at a constant rate (~0.2 ml/min) with extracellular solution containing 145 mM NaCl, 5 mM CsCl, 2 mM $CaCl_2$, 1 mM $MgCl_2$, 5 mM glucose, 5 mM sucrose, 10 mM Hepes (pH = 7.4 with NaOH), and intracellular solution containing 5 mM CsCl, 120 mM CsF, 10 mM NaCl, 11 mM EGTA, 1 mM $CaCl_2$, 1 mM $MgCl_2$, 2 mM

$Na_2ATP$, 10 mM Hepes (pH = 7.3 with CsOH). Sodium currents were recorded using an Axopatch 200B amplifier (Molecular Devices, Sunnyvale, CA) controlled by a pCLAMP 10/DigiData 1440 acquisition system (Molecular Devices). Patch electrodes were pulled from borosilicate glass capillaries (GC150TF-7.5, Harvard Apparatus Ltd.). Patch electrodes exhibited resistance values of $1.1-1.5$ MΩ; series resistance values, typically of $2-2.5$ MΩ, were 85–90% compensated. Currents and potentials were low-pass filtered at 10 kHz and digitized at 50 kHz. CHO cells of $12-25$ pF cell capacitance and expressing $2-10$ nA peak $Na_v1.2$ currents were included in data analysis; cells with small peak amplitude sodium currents (<2 nA, ~20% of cells) or large peak amplitude sodium currents (>10 nA, ~20%) were excluded[7]. Leak and capacitive currents were corrected using $-P/4$ subtraction, except when determining the steady-state inactivation and the recovery from fast inactivation. The voltage protocols are depicted in the figures and explained in the Results section. The current–voltage $(I-V)$ relationships were determined by measuring peak sodium current amplitudes against depolarizing voltage steps in the voltage range between $-80$ to $+60$ mV. Conductance $(G)$ was determined according to the equation $G = I/(V - V_{rev})$, where $V_{rev}$ is the $Na^+$ reversal potential. The normalized conductance values $(G/G_{max})$ were plotted against the membrane potential resulting in activation curves. The voltage dependence of steady-state fast inactivation was determined using 100-ms voltage pre-pulses in 5 mV increments, followed by a 10-ms voltage step to $-10$ mV to test the availability of the sodium current. Activation and inactivation curves were fit using the Boltzmann equation:

$$\frac{G}{G_{max}} = \frac{1}{[1 + e^{(V - V_{0.5})/k}]},$$ (1)

where $V$ is the membrane voltage, $V_{0.5}$ represents the voltage for half-maximal activation or inactivation ($V_{0.5,act}$ or $V_{0.5,inact}$, respectively), and $k$ is the slope factor. Persistent sodium current amplitude was measured 40 ms after the onset of the depolarizing voltage and represents the remaining inward current after $-P/4$ leak correction[7]. The time constants of sodium current inactivation were determined by fitting the decaying phase of the sodium current with a double-exponential equation:

$$\frac{I}{I_{max}} = A_f e^{-t/\tau_f} + A_s e^{-t/\tau_s},$$ (2)

where $t$ is time, $A_f$ and $A_s$ are the fractions of the fast and slow inactivation components, and $\tau_f$ and $\tau_s$ are the time constants of the fast and slow inactivating components, respectively. Recovery from fast inactivation was assessed using a paired-pulse voltage protocol from a holding potential of $-120$ mV. The first pulse, which inactivated the channels, was followed by a second pulse to measure the current fraction recovered from inactivation following inter-pulse intervals of increasing duration. The time constants of recovery $(\tau)$ were estimated by fitting a single exponential function to the data, as follows:

$$\frac{I}{I_{max}} = 1 - e^{-t/\tau},$$ (3)

where $t$ is the time between the P1 and P2 test pulses.

**Clinico-electrophysiological severity score of $Na_v1.2$ variants**. Thirteen early-infantile $Na_v1.2$ variants were allocated into benign, variable, and severe phenotypic groups (Supplementary Table 4). The clinico-electrophysiological severity score of $Na_v1.2$ variants (CESSNa+ score)[15] was calculated by quantifying changes of selected biophysical properties (i.e., $V_{0.5,act}$ and $V_{0.5,inact}$) relative to control (wild-type $Na_v1.2$ channel). For each assessed property, the maximum effect of all 13 variants was determined relative to control and divided in three thirds corresponding to large, medium, and small changes of a feature, associated with high (3), medium (2) and low (1) scores[15]. Additional points were scored for the maximal (strongest) change of a selected biophysical property (e.g., persistent sodium current and recovery from fast inactivation) and if multiple (at least three) biophysical properties were affected by a mutation for a given variant. Opposing effect, such as the shifts of the $V_{0.5,act}$ and $V_{0.5,inact}$ in the same direction were considered as 'neutralizing'. Higher CESSNa+ scores indicate greater derangement of biophysical properties compared with the wild-type channel, and have been previously correlated with pronounced clinical severity. Two sets of CESSNa+ scores were generated, by considering or omitting the neutralizing effect. Mean CESSNa+ scores in the benign, variable, and severe phenotypic groups were compared using one-way ANOVA and the $P$ value determined.

**Dynamic action potential clamp (DAPC) experiments**. Real-time DAPC recordings were performed using scaled sodium currents of individual $Na_v1.2$ channel variants implemented in a biophysically realistic axon initial segment (AIS) model[7]. The virtual sodium conductance of the AIS model was set to zero ($gNa_v1.6 = 0$), whereas the $K_v$ channel conductance value was set to default ($gK_v = 1$)[39] or $gK_v = 2$ (twice the original $gK_v$). Firing of the axon initial segment (AIS) compartment model was elicited using either step current injections in 2-pA increments, or synaptic stimuli of various intensity, as the sum of two independent excitatory and inhibitory synaptic conductances ($g_e$ and $g_i$, respectively)[7]. In experiments where the hybrid neuron's firing was initiated by scaled $g_e/g_i$ conductance ratios, the $gK_v = 1$ setting was used. We have previously demonstrated that $gK_v$ scaling had negligible effect on the AIS model firing activity upon synaptic noise stimulation[7]. Firing responses of hybrid neurons incorporating pathogenic $Na_v1.2$ variants were compared to those with wild-type, with increased firing considered GoF, and decreased firing LoF. The wild-type or mutant (external input) sodium current was adjusted (scaled) to 380–400 pA, hence differences in firing activity are mainly attributed to the altered biophysical properties of the given $Na_v1.2$ channel variant. Input–output relationships were determined by plotting action potential firing frequency against the amplitude of the stimulus current or against $g_e:g_i$ ratios. In all DAPC experiments, the original (non-scaled) external $I_{Nav1.2}$, the scaled external $I_{Nav1.2}$, the membrane potential ($V_m$), $g_e$, $g_i$, $gK_v$, and $gNa_v1.6$ were simultaneously recorded.

**Action potential frequency and morphology in DAPC experiments**. The firing frequency, rheobase, upstroke velocity, amplitude, width, and decay time were evaluated using the Clampfit module of pCLAMP 10, whereas the threshold was calculated in Axograph X (Axograph Scientific, Sydney, Australia). Firing frequency (in Hz) during step current injections or in the presence of scaled synaptic current was calculated as the number of action potentials per 1 s. The first action potential elicited by a current step 2 pA above rheobase was selected for action potential morphology analysis. Rheobase (in pA) was determined as the lowest value of injected current that yielded at least one action potential. Threshold (in mV) was defined as the point on the action potential rising phase where the first derivative ($dV/dt$) of the voltage trajectory reached 20 mV/ms. Upstroke velocity (in $dV/dt$) was defined as the maximum value of the first derivative of the action potential waveform. Amplitude (in mV) was determined as the most depolarized value of the action potential relative to the baseline (mean membrane voltage in the absence of firing). Width (in ms) was measured as the time between the half-amplitude points of the rising and decaying phases of the action potential. Decay time (in ms) is the time between the trace's crossing 90% and 10% of the baseline-to-peak amplitude range in the decay stage of the action potential.

**Statistics and reproducibility**. Correlations between genotype, biophysical profile, and phenotype were examined for all recurrent variants. Electrophysiological data were analyzed in Clampfit 9.2 (Molecular Devices) and plotted in Origin 2022 (Microcal Software Inc., Northampton, MA, USA). Mean and standard error of the mean (SEM) were used to describe the variability within groups; $n$ values represents the number of independent experiments. One-way ANOVA or two-way ANOVA followed by Dunnetts' post hoc test were performed for the statistical evaluation of data in GraphPad Prism 9.0 (La Jolla, CA, USA). Sample sizes and $P$ values are included in figure legends and in the Supplementary tables. Differences between groups were considered statistically significant if $P < 0.05$.

**3D structural modeling of $Na_v1.2$ variants**. In silico pathogenic variants of the human $Na_v1.2$ channel in the absence or presence of the auxiliary subunit $\beta_2$ (PDB accession no. 6J8E)[24] were made and visualized in PyMOL 2.3.2. software (Schroedinger LLC, New York, USA).

**Reporting summary**. Further information on research design is available in the Nature Research Reporting Summary linked to this article.

## Data availability
All data used for generating the main figures can be found in Supplementary Data 3. The other data will be available from the corresponding author upon reasonable request.

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

## Acknowledgements

We are grateful to all patients and their families participating in this research. We acknowledge the Simons Foundation/Simons Searchlight database in providing data for this study. We are grateful to the Simons Searchlight families, as well as the Simons VIP (Simons Searchlight) Consortium. We appreciate obtaining access to phenotypic and genetic data on SFARI Base. Approved researchers can obtain the Simons Searchlight population dataset described in this study by applying at https://base.sfari.org. This study was supported by an Australian Research Council Centre of Excellence for Integrative Brain Function grant (CE14010007), National Health and Medical Research Council (NHMRC) program grant (10915693) to S.P., S.F.B., and I.E.S, Medical Research Future Fund Genomic Health Futures Mission Project Grant to G.B., K.B.H., S.F.B., I.E.S., and S.P., and project funding to the *SCN2A* Natural History study by RogCon, Inc. and Praxis Precision Medicines to K.B.H. S.P. is supported by an NHMRC Fellowship, I.E.S. by an NHMRC Practitioner Fellowship and Investigator Fellowship, and K.B.H. by an NHMRC Early Career Fellowship and a clinician-scientist fellowship from the Murdoch Children's Research Institute (MCRI). G.B. was partly funded by RogCon, Inc. J.H. is supported by an Australian Government Research Training Program stipend and The Florey Institute of Neuroscience and Mental Health. The Florey Institute of Neuroscience and Mental Health and MCRI are supported by a Victorian State Government Operational Infrastructure Support Program.

## Author contributions

G.B. designed and carried out the electrophysiological studies and 3D structural modeling, and analyzed and interpreted data. K.B.H. designed and carried out the clinical studies and analyzed and interpreted clinical data. J.H. analyzed clinical data and compiled the Supplementary Data 2 file. N.O. carried out 3D structural modeling and interpreted data. J.R. and I.O. collected and analyzed clinical data. D.R.V., T.W., S.A.-H., G.L., M.A., P.V., I.E.S., S.F.B., and M.W. collected and analyzed clinical data. S.P. conceived and designed the electrophysiological studies. G.B., K.B.H., and S.P. wrote the manuscript. All authors contributed to reviewing and revising the manuscript.

## Competing interests

S.P. is co-founder and equity holder in Praxis Precision Medicines, Inc., Cambridge, MA, USA and RogCon, Inc., which develops precision medicines for neurogenetic disorders, including those caused by *SCN2A* mutations. S.P. is a Scientific Advisor and equity holder in Pairnomix, Inc., Minneapolis, MI, which is undertaking precision medicine development in epilepsy and related disorders. The remaining authors declare no competing interests.
