## [Peer Review File · Communications Biology]

Reviewers' comments:

Reviewer #1 (Remarks to the Author):

Comments

The manuscript by Berecki et al assesses the clinical phenotype of 179 patients with 38 recurrent variants in the SCN2A gene and characterizes the functional impact of 13 variants using whole-cell voltage clamp and dynamic action potential clamp. Many variants in the SCN2A gene associated with epilepsy show mixed gain- and loss-of-function effects when characterized using traditional voltage clamp recording, and predicting how those mixed effects impact neuronal function and epilepsy severity is difficult. This manuscript presents a compelling case that dynamic action potential clamp may be a better predictor of how these mutations alter neuronal excitability than voltage clamp, however, there are some concerns about the manuscript that should be addressed. Major Comments

1. While the authors have convincingly shown that DAPC can predict gain- and loss-of-function effects on excitability, they do not provide any statistical analyses of these experiments. While the authors state that they could not discern differences in predicted severity using DAPC, is this because this portion of the study was underpowered to do so? If the differences in excitability between severe and benign are subtle, then the 4-10 replicates studies for variants studied may not be enough to support this conclusion, and with no statistical analyses presented, it is difficult to make any conclusion.
2. The DAPC studies are performed by essentially modeling a homozygous state, as only the mutant allele is included in the "neuron". Do the results change if the heterozygous state is studied, by including a computational Nav1.2 WT allele?
3. As it is currently presented, the structural modeling distracts from the overall message of the manuscript. The authors do not go into any detailed discussion of any variant other than K905N, and predict that this variant may destabilize the Na1.2 alpha subunit with the Beta2 subunit. This section would be more powerful if the authors test this hypothesis, and record K905N in voltage clamp and DAPC mode in the presence of Beta2.
4. The authors state that several variants show "apparently decreased rheobase relative wild-type". Can this be explicitly measured using this DAPC? Also, can other aspects of action potential morphology, such as AP width and height, and AP phase plots to assess AP speed be measured to further assess the differences between predicted severity?

Minor points

1. The full name for the acronym S(F)NIS should be spelled out.
2. Page 6, line 127. Data from K908E are not presented. I believe this should be E999K.

Reviewer #2 (Remarks to the Author):

This study reports a comprehensive analysis of disease-associated SCN2A mutations. In addition to phenotypic data, the authors present biophysical data to assess for a correlation between biophysics and phenotype, and suggest that dynamic current clamp is a superior means for biophysical analyses. The bottom line appears to be that even with this large data set, challenges remain to the goal of predicting severity and/or GoF vs. LoF. Thus, the promise of precision medicine for SCN2A variants remains distant.

While the sheer amount of data presented are helpful, the manuscript is poorly organized and there appear to be multiple errors. Perhaps some of these are simply because of the poor organization, but the manuscript would be markedly improved with better organization and thorough checking of the data. As written, the manuscript is difficult to follow and the conclusions seem arbitrary.

For example, the tabulation of the variants appears to have some inaccuracies:

1. V1325L (should be V1325I?) is presented as one of the 6 recurrent variants for study, but it is

not in Table 1. R1629L is presented as one of the 11 recurrent variants for study, but it is not in Table 1. R1629L is separately reported as a non-recurrent variant (line 136).

2. K905N is reported as a recurrent variant in the pore module (line 133) and a non-recurrent variant (line 136).

3. D195G was studied with patch clamp, but not listed in the results introductory paragraph or the 11 recurrent variants chosen for biophysical analysis.

4. Of the variants chosen for biophysical study, only one (E1211K) is listed as a LoF variant, to the conclusions re: LoF variants cannot be generalized.

5. The definition of phenotypic severity, buried in a supplemental Table, would be more helpful when first discussed in results (line 162). Or at least please put a reference to the supplemental Table at this place.

Proofreading text would also improve the manuscript. For example:

1. Abstract, line 53: "Here we classify clinical phenotypeS of 179 individuals with 38 recurrent SCN2A variants..."

2. Intro, line 6: "Mutations in the SCN2A gene encoding the voltage-gated sodium channel Nav1.2, ..." eliminate the comma.

Another "flaw" is that of the multiple variants studied, they are almost exclusively LoF, so generalizing to LoF variants is challenging.

Other suggestions:

Figure 2/supplementary Fig 2: the examples traces do not have scale bars.

RESPONSE TO REVIEWERS

Reviewer #1 (Remarks to the Author):

Comments

The manuscript by Berecki et al assesses the clinical phenotype of 179 patients with 38 recurrent variants in the SCN2A gene and characterizes the functional impact of 13 variants using whole-cell voltage clamp and dynamic action potential clamp. Many variants in the SCN2A gene associated with epilepsy show mixed gain- and loss-of-function effects when characterized using traditional voltage clamp recording, and predicting how those mixed effects impact neuronal function and epilepsy severity is difficult. This manuscript presents a compelling case that dynamic action potential clamp may be a better predictor of how these mutations alter neuronal excitability than voltage clamp, however, there are some concerns about the manuscript that should be addressed.

Major Comments

1. While the authors have convincingly shown that DAPC can predict gain- and loss-of-function effects on excitability, they do not provide any statistical analyses of these experiments.

We apologise for not providing proper description of the statistical analyses for DAPC experiments in the main text. In the revised manuscript, we reanalysed the firing activity of the hybrid neuron during step current stimulation or during synaptic current stimulation. We specified the statistical analyses in the text and figure legends, and changed the figures showing DAPC data to highlight the statistically significant differences between various datasets as follows:

Supplementary Tables 6 and 8 show the mean firing frequency \pm standard error of the mean (SEM) values and the results of the statistical evaluation with probability (P) values from two-way ANOVA. We also performed statistical analyses of the action potential morphology in DAPC experiments and show the results in a new Supplementary Table 7. The mean \pm SEM values of the biophysical properties of the wild-type or K905N Na_v1.2 channel variants co-expressed with β_2 subunit in voltage clamp experiments are shown in Supplementary Table 10 (these results are discussed in more detail below in our response to comment #4). In Supplementary Tables 11 and 12, we included the statistical evaluation of hybrid neuron excitability and action potential morphology, respectively, with wild-type or early-infantile K905N variants co-expressed with β_2 subunit in DAPC experiments.

In the legend of Figure 2 (revised manuscript), we added the following sentence: “*P < 0.05, one-way ANOVA, followed by Dunnett’s post-hoc test. See the detailed statistical evaluation of biophysical characteristics of the variants in Supplementary Table S4”. In this sentence, we now correctly refer to Supplementary Table 4 (deleted ‘Table 5’).

In Figure 3B (revised manuscript), by adding asterisks to highlight statistically significant differences in firing activity relative to wild-type. In the legend of this figure, we added details of the statistical analysis as follows: “Firing frequencies relative to WT were assessed using two-way ANOVA followed by Dunnett’s post-hoc test; asterisks indicate P < 0.05 (for individual P values see Supplementary Table 6). Note the decreased or increased rheobase with the EI-severe R1629L and later-onset D195G variants, respectively. The statistical evaluation of the action potential morphology is shown in Supplementary Table 7.” Our responses regarding the analysis of the rheobase and other aspects of action potential morphology are included in the response to comment #4 below.

In Figure 4B (revised manuscript), we added asterisks to highlight the differences in firing activity relative to wild-type. In the legend, we added a sentence detailing the statistical analysis and slightly re-worded the second last sentence as follows: “Two-way ANOVA, followed by Dunnett’s post-hoc test, was used to compare the AP firing frequencies elicited by scaled excitatory to inhibitory conductance ratios (g_e/g_i) in the

presence of Nav1.2 variants; asterisks indicate $P < 0.05$ (see individual P values in Supplementary Table 8). Note the increased firing activity in the early-infantile severe/variable groups compared to wild-type, whereas later-onset variants result in an almost complete loss of firing.”

1. (continued) While the authors state that they could not discern differences in predicted severity using DAPC, is this because this portion of the study was underpowered to do so? If the differences in excitability between severe and benign are subtle, then the 4-10 replicates studies for variants studied may not be enough to support this conclusion, and with no statistical analyses presented, it is difficult to make any conclusion.

We performed our DAPC assay in two different stimulation modes (using step currents or synaptic conductance), with similar outcomes. Our statistical analyses convincingly validate the differences in firing activities with the variants relative to wild-type. These analyses are presented in the revised manuscript.

One of the major findings of our study was that the phenotypic severity of patients with recurrent *SCN2A* variants varies between individuals with the same variant for 40% of variants associated with early-infantile phenotypes. For example, in Table 1 we show that individuals with the same recurrent variant may display variable clinical severity, ranging from unaffected to severe (e.g., the A263V and R1319Q variants have been seen in individuals whose early infantile phenotypes range from ‘unaffected’, ‘benign’, ‘intermediate’, to ‘severe’). While DAPC evaluation efficiently predicts the GoF characteristics of these variants compared with wild-type, this highlights that biophysical properties of such recurrent variants cannot be the sole determinant of phenotypic severity. In the Discussion, we emphasised that clinical phenotype variability among patients carrying identical (recurrent) mutations “appears more common than previous reports have suggested, confirming that the variant itself is an unreliable predictor of outcome severity”.

Further, our data also convincingly demonstrate that neither DAPC/VC, nor the CESSNa⁺ score, generated from the analysis of biophysical parameters of individual variants, could reliably segregate by severity the early-infantile variants that were consistently associated with only a ‘benign’ or only a ‘severe’ phenotype.

In the last paragraph of the Discussion, we originally included that “Factors influencing severity are not clear and were not studied here” and mentioned that various secondary or compensatory mechanisms might affect severity.

2. The DAPC studies are performed by essentially modeling a homozygous state, as only the mutant allele is included in the “neuron”. Do the results change if the heterozygous state is studied, by including a computational Nav1.2 WT allele?

Thank you for this suggestion. It would have been interesting to explore the impact of specific ion channel mutations in neuron models of various complexity. In this manuscript, we were focussed more on the performance of DAPC vs VC for reliably and efficiently predicting the functional impact of a variant (and whether the predictions correlated with phenotypic group and severity) and have considered this beyond the scope of the current study. In our previous studies^{1,2} and in this manuscript, we tested the sensitivity of the hybrid neuron by scaling the virtual potassium and/or sodium conductances (gK and/or gNav1.6) in the axon initial segment (AIS) compartment model. We agree that assessing the behavior of a model cell mimicking Na_v1.2 channel heterozygosity is an important question however, and plan to address it by incorporating virtual Na_v1.2 conductance into our AIS model in future experiments and a follow-up paper.

3. As it is currently presented, the structural modeling distracts from the overall message of the manuscript. The authors do not go into any detailed discussion of any variant other than K905N, and predict that this variant may destabilize the Na1.2 alpha subunit with the Beta2 subunit.

Thank you for the suggestion to add details of the structural modeling of all variants in the manuscript. In the ‘3D structural modeling’ subchapter (Results), we reworded the first two sentences and extended the first paragraph by adding brief descriptions of the predicted structural changes for all variants to better highlight the relevance of the structural model supporting functional data as follows: “To better understand the molecular basis for the functional data from VC and DAPC assays, we mapped the variants onto a 3D model

of the Nav1.2 channel³. The detailed views of the channel, the predicted structural changes due to the mutated residues, and the interpretations of the effects caused by these modifications are shown in Supplementary Figure 7. Briefly, the D195G mutation disrupts polar interactions in S3_{DI}; V261L and A263V mutations affect hydrophobic interactions in DI; Q383E affects the key E384 residue in the ‘DEKA’ selectivity filter⁴; R856Q, R1319Q, and R1629L affect gating-charge carrying R residues, which are key for voltage sensor movements; E1211K involves the change of a highly conserved negative residue to positive in S4_{DIII}; E1321K and V1325I affect coupling interactions between the voltage sensor and the pore⁵, and residues involved in fast inactivation⁶ in S4-5_{DIII}; and Q1531K results in the change of a conserved residue with polar uncharged side chain to a positive residue in S1_{DIV}. The structure of the DII-DIII linker carrying the E999K mutation is currently not resolved.” Further, in the second paragraph, we re-worded the “Assessment of the 3D structure of the wild-type Nav1.2 α_1 subunit in complex with the β_2 subunit”... sentence.

3. (continued) This section would be more powerful if the authors test this hypothesis, and record K905N in voltage clamp and DAPC mode in the presence of Beta2.

Thank you for the suggestion to evaluate the impact of K905N variant in the presence of β_2 subunit. We co-expressed the wild-type or the K905N α_1 pore-forming subunit and the β_2 subunit and performed a series of experiments in VC or DAPC modes, which brought out the biophysical impact and confirmed the expected GoF.

In the revised manuscript, these experiments are included in the Result section and shown in a new figure and new tables (Figure 5, Supplementary Tables 10-12). Overall, we were able to demonstrate that the K905N mutation destabilizes the interaction between the pore-forming α_1 subunit and β_2 , resulting in GoF compared with wild-type α_1 subunit and β_2 .

We added the new subchapter ‘Predicting the functional impact of K905N variant using co-expression of α_1 and β_2 subunits’ to the Results, as follows: “We hypothesised that the K905N mutation indirectly destabilized key electrostatic interactions in S5_{DII} and/or between S5_{DII} and the β_2 subunit (Fig. 5B). To test the impact of β_2 on Nav1.2 channel variant function, we co-expressed the wild-type or the K905N α_1 pore-forming subunit and the β_2 subunit (WT + β_2 and K905N + β_2 , respectively) in CHO cells. Simultaneously, we also repeated the experiments with CHO cells transfected with wild-type α_1 subunit alone (WT*) to enhance experimental control. In VC experiments, the activation, inactivation, and recovery from inactivation parameters for WT* channels (Supplementary Table 9) were indistinguishable from those of the wild-type channels shown in Figure 2, Supplementary Figure 5, and Supplementary Table 4. Co-expression of the β_2 subunit did not affect the current density of the assessed variants (Supplementary Table 10) but shifted the voltage dependence of the wild-type and K905N variants (Supplementary Table 10). The depolarizing shift of the WT + β_2 activation curve (Fig. 5C) agrees with published data⁷. Relative to WT + β_2 channels, the activation and inactivation curves of K905N + β_2 exhibited hyperpolarizing and depolarizing shifts, respectively, which correspond to gain-of-function (Fig. 5C), whereas the time course of recovery from fast inactivation for K905N + β_2 was unchanged (Fig. 5D and Supplementary Table 10). In DAPC mode, the hybrid cell model incorporating K905N + β_2 or WT* current achieved significantly higher firing frequencies over a range of step stimuli relative to WT + β_2 (Fig. 5E and Supplementary Table 11). These hybrid cells exhibited similar action potential characteristics except the decreased threshold of K905N + β_2 relative to WT + β_2 (Supplementary Table 12).

Our data suggest that the decreased excitability of the hybrid neuron incorporating WT + β_2 current relative to WT* is due to the interaction between the heterologously expressed α_1 and β_2 subunits; the K905N mutation hinders these interactions, resulting in GoF.”

We also reworded the Abstract by adding the following sentences: “The functional impact of the one variant not resolved by either DAPC or VC, was brought out by co-expression of the α_1 and β_2 subunits of the Nav1.2 channel, after 3D molecular modeling suggested the variant may impact interactions between these subunits. Despite strong correlation between biophysical impact and phenotypic group, biophysical testing was not suitable to reliably segregate early-infantile variants by severity.”

In the revised Methods, we described the co-transfections involving the β_2 subunit by adding the following sentence: “Co-transfections involving the β_2 subunit were performed using 4 μg wild-type or K905N sodium channel α_1 subunit, 3 μg human β_2 subunit (NCBI Reference Sequence: NM_004588.5; Origene Technologies, Rockville, MD) and 1 μg eGFP.”

In the revised Discussion, we added the following paragraph in the ‘Determining biophysical impact’ subchapter: “Previous studies have demonstrated that the β_2 subunit exerts depolarizing effects on $\text{Na}_v1.2$ gating and alters cell surface expression of $\text{Na}_v1.2$ ^{7,8}, and our molecular modeling suggested the K905N variant may alter α_1 and β_2 subunit interactions. We therefore hypothesised that co-expression of the α_1 and β_2 subunit interactions would be needed to reveal the impact of this mutation; our results support this hypothesis, revealing a GoF, consistent with that expected for the phenotype. This finding highlights that molecular modeling and subunit co-expression may be useful tools in determining variant impact for the minority of cases in which this is not resolved by DAPC.”

4. The authors state that several variants show “apparently decreased rheobase relative to wild-type”. Can this be explicitly measured using this DAPC? Also, can other aspects of action potential morphology, such as AP width and height, and AP phase plots to assess AP speed be measured to further assess the differences between predicted severity?

Thank you for the suggestion to determine the contribution of wild-type and mutated channels to the action potential morphology - DAPC data can efficiently reveal action potential characteristics.

In the revised Supplementary Methods, we added the ‘Action potential frequency and morphology in DAPC experiments’, subchapter as follows: “The firing frequency, rheobase, upstroke velocity, amplitude, width, and decay time were evaluated using the Clampfit module of pCLAMP 10, whereas the threshold was calculated in Axograph X (Axograph Scientific, Sydney, Australia). Firing frequency (in Hz) during step current injections or in the presence of scaled synaptic current was calculated as the number of action potentials per 1 s. The first action potential elicited by a current step 2 pA above rheobase was selected for action potential morphology analysis. Rheobase (in pA) was determined as the lowest value of injected current that yielded at least one action potential. Threshold (in mV) was defined as the point on the action potential rising phase where the first derivative (dV/dt) of the voltage trajectory reached 20 mV/ms. Upstroke velocity (in dV/dt) was defined as the maximum value of the first derivative of the action potential waveform. Amplitude (in mV) was determined as the most depolarized value of the action potential relative to the baseline (mean membrane voltage in the absence of firing). Width (in ms) was measured as the time between the half-amplitude points of the rising and decaying phases of the action potential. Decay time (in ms) is the time between the trace's crossing 90% and 10% of the baseline-to-peak amplitude range in the decay stage of the action potential.

In the new Supplementary Table 7, we included the mean \pm SEM values of the action potential parameters and their statistical evaluation.

In the revised manuscript, we summarized the results of this analysis in the ‘Predicting the effects of SCN2A variants on action potential firing using DAPC approach’ subchapter, as follows: “We assessed the contribution of $\text{Na}_v1.2$ variants to the action potential morphology by determining the rheobase, threshold, upstroke velocity, amplitude, width, and decay time. Action potential characteristics of several variants were altered relative to wild-type (Supplementary Table 7); the increased width and decay time for E999K, R856Q, A263V, V1325I, Q383, and Q1531K was correlated with the increased $I_{\text{Na-P}}$ of these variants in VC mode (Supplementary Figure 3).”

In the legend of Figure 3 (revised manuscript), we added the following sentence: “Note the decreased or increased rheobase with the EI-severe R1629L and later-onset D195G variants, respectively. The statistical evaluation of the action potential morphology is shown in Supplementary Table 7.” We recognize that rheobase analysis could have been more accurate if smaller increments for the injected currents were used at/around rheobase.

Minor points

1. The full name for the acronym S(F)NIS should be spelled out.

Thank you for pointing this out. In the revised Introduction, we spelled out the first in-text reference to S(F)NIS as ‘self-limited (familial) neonatal-infantile seizures’. We also spelled out S(F)NIS in the Glossary.

2. Page 6, line 127. Data from K908E are not presented. I believe this should be E999K.

“Six variants arose in both *de novo* and inherited forms (A263V, K908E, R1319Q, E1321K, V1325I, Q1531K).” This sentence describing the ‘*de novo* and inherited’ variants is correct (‘Individuals and variants’ subchapter in Results). Patient data of the K908E variants are included in the Supplementary data 1 Table ‘Phenotypic data of 179 individuals with 38 recurrent SCN2A variants, and 2 individuals with unique variants.’

Reviewer #2 (Remarks to the Author):

This study reports a comprehensive analysis of disease-associated SCN2A mutations. In addition to phenotypic data, the authors present biophysical data to assess for a correlation between biophysics and phenotype, and suggest that dynamic current clamp is a superior means for biophysical analyses. The bottom line appears to be that even with this large data set, challenges remain to the goal of predicting severity and/or GoF vs. LoF. Thus, the promise of precision medicine for SCN2A variants remains distant.

While the sheer amount of data presented are helpful, the manuscript is poorly organized and there appear to be multiple errors. Perhaps some of these are simply because of the poor organization, but the manuscript would be markedly improved with better organization and thorough checking of the data. As written, the manuscript is difficult to follow and the conclusions seem arbitrary.

Comments: the tabulation of the variants appears to have some inaccuracies:

Thank you for pointing this out. We apologise for the inaccuracies with indexing/tabulating of the variants. In the revised version of the manuscript, we have re-organized several sections and paragraphs, and have corrected the errors and typos. The changes undertaken are listed below:

1a: V1325L (should be V1325I?) is presented as one of the 6 recurrent variants for study, but it is not in Table 1.

We apologize for the mistake. In the ‘Individuals and variants’ subchapter (Results), we corrected the sentence including the V1325I variant as follows: “Six variants arose in both *de novo* and inherited forms (A263V, K908E, R1319Q, E1321K, V1325I, Q1531K).” Table 1 shows that the clinical phenotype of patients carrying the recurrent V1325I variant can be ‘early-infantile benign’ or ‘early-infantile severe’.

1b: R1629L is presented as one of the 11 recurrent variants for study, but it is not in Table 1. R1629L is separately reported as a non-recurrent variant (line 136).

Thank you for pointing out that the text referring to R1629L needs clarification. We appreciate that discussing the S4 segment localization of non-recurrent R1629L, together with the recurrent R856Q and R1319Q variants was confusing. We re-worded the ‘Individuals and variants’ subchapter (Results) as follows: “Eleven recurrent variants were chosen for biophysical analysis, spanning both early-infantile and later-onset phenotypes, and seen in individuals with a range of severities. To further understand the relationships between electrophysiological findings and severe phenotypes, we also studied biophysically two non-recurrent variants, K905N and R1629L, associated with severe early-infantile phenotypes. These two individuals are not included in the analysis of phenotypic data.”

R1629L is a non-recurrent variant, therefore is not included in Table 1. The original sentence discussing the localization of the mutations was moved in the ‘Biophysical characterization of Nav1.2 channel variants using VC recordings’ subchapter (Results).

2. K905N is reported as a recurrent variant in the pore module (line 133) and a non-recurrent variant (line 136).

The K905N variant is non-recurrent. We clarified its status in the same way as for the R1629L variant above. Like R1629L, the K905N variant is not included in Table 1.

3. D195G was studied with patch clamp, but not listed in the results introductory paragraph or the 11 recurrent variants chosen for biophysical analysis.

In the revised manuscript, the results introductory paragraph was moved in the ‘Biophysical characterization of Nav1.2 channel variants using VC recordings’ subchapter (Results). This paragraph was reworded to include a precise description of the localization of all variants studied biophysically. In addition to D195G, we also included the description of the localization for E1211K, Q1531K, and E999K.

The revised sentence reads as follows: “The variants are localized in channel regions associated with specific functions, including voltage sensing: R856Q, R1319Q, and R1629L (in segment 4 of domain II (S4_{DII}), S4_{DIII}, and S4_{DIV}, respectively); channel gating: E1211K, Q1531K, and D195G (in S1_{DIII}, S1_{DIV}, and S3_{DI}, respectively); fast inactivation: E1321K and V1325I (in the S4-S5_{DIII} linker specifically implicated in forming the inactivation gate receptor⁹); pore module⁶: A263V, V261L (both in S5_{DI}), K905N (S5_{DII}), and Q383E (S5-S6_{DI} turret loop:); and protein trafficking to the axon initial segment (AIS)¹⁰: E999K (DII-DIII linker) (Figure 1).”

4. Of the variants chosen for biophysical study, only one (E1211K) is listed as a LoF variant, to the conclusions re: LoF variants cannot be generalized.

Literature data suggests that LoF missense variants are mainly seen in later-onset infantile epilepsy presenting after 3 months of age^{11,12}. Our study included two loss-of-function (LoF) variants: E1211K and D195G. In the ‘Biophysical characterization of Nav1.2 channel variants using VC recordings’ subchapter (Results), we have defined that “The 13 variants studied (Figure 1) included 11 associated with an early-infantile phenotype and two with a later-onset phenotype (E1211K, D195G).”

Both the E1211K and the D195G variant resulted in LoF in dynamic action potential clamp (DAPC) experiments. Our previous DAPC study has demonstrated that the recurrent R853Q variant, associated with later-onset clinical phenotype, also results in LoF¹. Nevertheless, we agree that more clinical and functional data is needed to establish definite associations between later-onset cases and LoF, and will address this in a comprehensive follow-up combined clinical and biophysical study of over 15 missense variants associated with later-onset epilepsy or ID/ASD to examine, identify, and validate associations between biophysical properties and clinical features including outcome severity in individuals with LoF variants.

In the Discussion chapter of the revised manuscript, we acknowledge that the small number of recurrent variants associated with later-onset epilepsies and ID/ASD represents one of the limitations in our study. The paragraph beginning “Our study included only a small number of recurrent variants associated with later-onset epilepsies and ID/ASD. These phenotypes are thought to arise because of LoF, as previously shown for some variants (e.g., R853Q), although conflicting results have been seen for other variants^{1,13-15}.” addresses this issue.

5. The definition of phenotypic severity, buried in a supplemental Table, would be more helpful when first discussed in results (line 162). Or at least please put a reference to the supplemental Table at this place.

Thank you for this suggestion. In the revised Methods (‘Clinical data’ subchapter), we provide the criteria of allocating the early-infantile patients according to severity and refer to Supplementary Table 2. In the

previous version of the manuscript, this table was erroneously labeled 3 instead of 2 (which is now corrected).

In the revised Results ('Early-infantile phenotypes in individuals with recurrent variants' subchapter), we reworded the sentence dealing with outcomes as follows: "Outcomes of individuals with early-infantile phenotypes were severe (n=35), intermediate (n=23) and benign (n=81), defined by criteria described in Methods and summarized in Supplementary Table 2"...

Proofreading text would also improve the manuscript. For example:

Thank you. We have proofread the revised manuscript and corrected the typos at various places in the text.

1. Abstract, line 53: "Here we classify clinical phenotypes of 179 individuals with 38 recurrent SCN2A variants..."

We have corrected 'phenotype' to 'phenotypes' in the Abstract.

2. Intro, line 6: "Mutations in the SCN2A gene encoding the voltage-gated sodium channel Nav1.2, ..." eliminate the comma.

As suggested, we removed the comma.

Another "flaw" is that of the multiple variants studied, they are almost exclusively LoF, so generalizing to LoF variants is challenging.

We believe that this comment is "of the multiple variants studied, they are almost exclusively LoF". Of the variants studied, there are fewer LoF than GoF. We agree that generalizing to LoF is challenging and, for this reason, focus the work on prediction of outcome severity on the GoF variants only. Please see also our discussion of the associations between later-onset cases and LoF in our response to comment #4.

Other suggestions:

Figure 2/supplementary Fig 2: the examples traces do not have scale bars.

We apologise for not explaining in full the scaling of current traces. In both Figure 2 and Supplementary Figure 2, current traces of individual variants were normalized to the same amplitude to facilitate direct comparison of the variants. We added the missing time scale bar to Figure 2. The time scale bar was already present in Supplementary Figure 2. In the legends of both figures we added the following sentence: "Current traces of individual variants were normalized to the same amplitude; note inset time scale bar."

Similarly, we included the above sentence in the legend of Supplementary Figure 3.

ADDITIONAL CLARIFICATIONS:

In addition to changes suggested by the reviewers, we made additional changes in the revised version of the manuscript as follows:

Methods: we reworded the 'Statistics and reproducibility' sub-chapter.

Results: In the 'Later-onset phenotypes and ID/ASD without epilepsy in individuals with recurrent variants' subchapter, we corrected labelling for L1342P. The corrected sentence reads as follows: "For five variants (D195G, R220Q, R853Q, E1211K, L1342P), all individuals (n=27) had a later-onset phenotype,"...

Discussion: in the last paragraph of the 'Determining biophysical impact' subchapter, we changed the sentence starting "A number of LoF recurrent variants"... to "Several LoF recurrent variants"...

Figure 2A: we added time scale bar inset. In the legend we added the following sentence in (A): “Current traces of individual variants were normalized to the same amplitude; note inset time scale bar.”

New references added to the list:

Catterall, W. A. From ionic currents to molecular mechanisms: the structure and function of voltage-gated sodium channels. *Neuron* **26**, 13-25, doi:10.1016/s0896-6273(00)81133-2 (2000).

Destexhe, A., Rudolph, M. & Pare, D. The high-conductance state of neocortical neurons in vivo. *Nature reviews. Neuroscience* **4**, 739-751, doi:10.1038/nrn1198 (2003).

Garrido, J. J. *et al.* Identification of an axonal determinant in the C-terminus of the sodium channel Na(v)1.2. *EMBO J* **20**, 5950-5961, doi:10.1093/emboj/20.21.5950 (2001).

Johnson, D. & Bennett, E. S. Isoform-specific effects of the beta2 subunit on voltage-gated sodium channel gating. *J Biol Chem* **281**, 25875-25881, doi:10.1074/jbc.M605060200 (2006).

In Supplementary Methods (‘Dynamic action potential clamp (DAPC) experiments’ subchapter), we added the following sentence: “Firing of the axon initial segment (AIS) compartment model was elicited using either step current injections in 2-pA increments, or synaptic current, generated as the sum of two independent excitatory and inhibitory synaptic conductances (ge and gi, respectively)¹.”

Supplementary Figure 2: we added the following sentence in the legend: “The membrane potential for half-maximal inactivation ($V_{0.5,inact}$) values and their statistical evaluation are shown in Supplementary Table 4.”

Supplementary Figure 3: we added the following sentence in the legend: “Current traces of individual variants were normalized to the same amplitude;...; note inset time scale bar.”

Supplementary Figure 5. We re-scaled the horizontal axes of the plots shown in the A panel to better highlight the differences between the recovery from fast inactivation curves of individual variants. In the legend, we added that “the statistical evaluation of τ values is shown in Supplementary Table 4”

Supplementary Figure 6. In panel B, we added asterisks to the graphs to indicate statistically significant differences between the firing frequencies of the Nav1.2 variants compared with wild-type. In the legend we added the following text: “Two-way ANOVA, followed by Dunnett’s post-hoc test, was used to compare the AP firing frequencies elicited by step stimuli in the presence of Nav1.2 variants; asterisks indicate $P < 0.05$;”

Supplementary Table 3: In the previous version of the manuscript, this table was erroneously labeled 2 instead of 3 (which is now corrected).

Supplementary Table 4: In the list of abbreviations, we added ‘VC, voltage clamp;’

Supplementary Table 6: We updated the statistical evaluation in the table using multiple comparisons in two-way ANOVA followed by Dunnett’s post-hoc test, and the P values (* $P < 0.05$, ** $P < 0.01$, *** $P < 0.001$, and **** $P < 0.0001$).

Supplementary Table 8: We updated the statistical evaluation in the table and added the following text below the table: “Statistically significant differences between the wild-type (WT) and mutant channels were determined using two-way ANOVA, followed by Dunnett’s post-hoc test (* $P < 0.05$, ** $P < 0.01$, *** $P < 0.001$, and **** $P < 0.0001$). # $P < 0.05$ at $g_e/g_i \geq 4$ (see Fig. 4B).”

Supplementary Table 9: We added a row to the ‘Early-infantile-severe group’ and reworded the text below the table to include the results with K905N variant co-expressed with β_2 subunit.

REFERENCES

- 1 Berecki, G. *et al.* Dynamic action potential clamp predicts functional separation in mild familial and severe de novo forms of SCN2A epilepsy. *Proc Natl Acad Sci U S A* **115**, E5516-E5525, doi:10.1073/pnas.1800077115 (2018).
- 2 Berecki, G. *et al.* SCN1A gain of function in early infantile encephalopathy. *Annals of neurology*, doi:10.1002/ana.25438 (2019).
- 3 Pan, X. *et al.* Molecular basis for pore blockade of human Na(+) channel Nav1.2 by the mu-conotoxin KIIIa. *Science* **363**, 1309-1313, doi:10.1126/science.aaw2999 (2019).
- 4 Syrbe, S. *et al.* Phenotypic Variability from Benign Infantile Epilepsy to Ohtahara Syndrome Associated with a Novel Mutation in SCN2A. *Mol Syndromol* **7**, 182-188, doi:10.1159/000447526 (2016).
- 5 Arcisio-Miranda, M., Muroi, Y., Chowdhury, S. & Chanda, B. Molecular mechanism of allosteric modification of voltage-dependent sodium channels by local anesthetics. *J Gen Physiol* **136**, 541-554, doi:10.1085/jgp.201010438 (2010).
- 6 Clairfeuille, T. *et al.* Structural basis of alpha-scorpion toxin action on Nav channels. *Science* **363**, doi:10.1126/science.aav8573 (2019).
- 7 Johnson, D. & Bennett, E. S. Isoform-specific effects of the beta2 subunit on voltage-gated sodium channel gating. *J Biol Chem* **281**, 25875-25881, doi:10.1074/jbc.M605060200 (2006).
- 8 Brackenbury, W. J. & Isom, L. L. Na Channel beta Subunits: Overachievers of the Ion Channel Family. *Front Pharmacol* **2**, 53, doi:10.3389/fphar.2011.00053 (2011).
- 9 Catterall, W. A. From ionic currents to molecular mechanisms: the structure and function of voltage-gated sodium channels. *Neuron* **26**, 13-25, doi:10.1016/s0896-6273(00)81133-2 (2000).
- 10 Garrido, J. J. *et al.* Identification of an axonal determinant in the C-terminus of the sodium channel Na(v)1.2. *EMBO J* **20**, 5950-5961, doi:10.1093/emboj/20.21.5950 (2001).
- 11 Wolff, M. *et al.* Genetic and phenotypic heterogeneity suggest therapeutic implications in SCN2A-related disorders. *Brain*, doi:10.1093/brain/awx054 (2017).
- 12 Wolff, M., Brunklaus, A. & Zuberi, S. M. Phenotypic spectrum and genetics of SCN2A-related disorders, treatment options, and outcomes in epilepsy and beyond. *Epilepsia* **60 Suppl 3**, S59-S67, doi:10.1111/epi.14935 (2019).
- 13 Begemann, A. *et al.* Further corroboration of distinct functional features in SCN2A variants causing intellectual disability or epileptic phenotypes. *Mol Med* **25**, 6, doi:10.1186/s10020-019-0073-6 (2019).
- 14 Ben-Shalom, R. *et al.* Opposing Effects on NaV1.2 Function Underlie Differences Between SCN2A Variants Observed in Individuals With Autism Spectrum Disorder or Infantile Seizures. *Biol Psychiatry* **82**, 224-232, doi:10.1016/j.biopsych.2017.01.009 (2017).
- 15 Mason, E. R. *et al.* Resurgent and Gating Pore Currents Induced by De Novo SCN2A Epilepsy Mutations. *eNeuro* **6**, doi:10.1523/ENEURO.0141-19.2019 (2019).

REVIEWERS' COMMENTS:

Reviewer #1 (Remarks to the Author):

The revised manuscript describing clinical phenotype of functional impact of recurrent SCN2A variants has added new data and clarified statistical analyses, resulting in a much stronger manuscript. They have addressed all concerns and I believe the manuscript acceptable for publication.

Reviewer #2 (Remarks to the Author):

The revision addresses the comments I previously shared, and offers an improved manuscript.